# Polyelectrolyte elastomer-based ionotronic sensors with multi-mode sensing capabilities via multi-material 3D printing

Caicong Li[1,2,4], Jianxiang Cheng[1,3,4], Yunfeng He [1,2,4], Xiangnan He[1,3], Ziyi Xu[1,2], Qi Ge [1,3] ✉ & Canhui Yang [1,2] ✉

Stretchable ionotronics have drawn increasing attention during the past decade, enabling myriad applications in engineering and biomedicine. However, existing ionotronic sensors suffer from limited sensing capabilities due to simple device structures and poor stability due to the leakage of ingredients. In this study, we rationally design and fabricate a plethora of architected leakage-free ionotronic sensors with multi-mode sensing capabilities, using DLP-based 3D printing and a polyelectrolyte elastomer. We synthesize a photo-polymerizable ionic monomer for the polyelectrolyte elastomer, which is stretchable, transparent, ionically conductive, thermally stable, and leakage-resistant. The printed sensors possess robust interfaces and extraordinary long-term stability. The multi-material 3D printing allows high flexibility in structural design, enabling the sensing of tension, compression, shear, and torsion, with on-demand tailorable sensitivities through elaborate programming of device architectures. Furthermore, we fabricate integrated ionotronic sensors that can perceive different mechanical stimuli simultaneously without mutual signal interferences. We demonstrate a sensing kit consisting of four shear sensors and one compressive sensor, and connect it to a remote-control system that is programmed to wirelessly control the flight of a drone. Multi-material 3D printing of leakage-free polyelectrolyte elastomers paves new avenues for manufacturing stretchable ionotronics by resolving the deficiencies of stability and functionalities simultaneously.

Ionotronics are a family of devices that function by hybridizing ions and electrons[1], enabling myriad applications as diverse as electro-physiological therapy[2], batteries[3], supercapacitors[4], fuel cells[5], and among others[6,7]. The advent and fast evolution of stretchable ionic conductors during the past decade has expanded the boundary of the field significantly, leading to stretchable ionotronics for tremendous applications that are previously inaccessible, such as artificial muscles[8], skins[9], axons[10], ionic touchpads[11], artificial eel[12], stretchable

ionotronic luminescence[13], ionotronic thermometry[14], and tough thermocells[15]. As the most quintessential stretchable ionic conductors, gel materials, including hydrogels[7] and ionogels[16], are advantageous in terms of intrinsic softness, stretchability, optical transparency, and biocompatibility. However, gels are suffering from inherent limitations of solvent leakage and/or evaporation, which inevitably deteriorate the performances of ionotronic devices[17]. The ionically conductive elastomers synthesized by dissolving lithium salt (e.g. lithium

[1]Shenzhen Key Laboratory of Soft Mechanics & Smart Manufacturing, Southern University of Science and Technology, Shenzhen, Guangdong 518055, P.R. China. [2]Soft Mechanics Laboratory, Department of Mechanics and Aerospace Engineering, Southern University of Science and Technology, Shenzhen, Guangdong 518055, P.R. China. [3]Department of Mechanical and Energy Engineering, Southern University of Science and Technology, Shenzhen, Guangdong 518055, P.R. China. [4]These authors contributed equally: Caicong Li, Jianxiang Cheng, Yunfeng He. ✉e-mail: geq@sustech.edu.cn; yangch@sustech.edu.cn

bis(trifluoromethanesulfonyl)imide (LiTFSI))[18] or zwitterions[19] into elastomer matrices are immune to solvent leakage, but they are still susceptible to ion leakage. The persistent concentration gradient between the interior and the exterior keeps driving the mobile ions to diffuse outwards when in contact with other elastomers. Polyelectrolyte elastomers with either cations or anions fixed to the backbones provide a promising remedy for endurant ionotronics by simultaneously resolving the predicaments of solvent-leakage and ion-leakage[17,20]. Nevertheless, in addition to the stability of materials, the manufacturing of devices has been another long-standing hurdle for the development of ionotronics.

Ionotronic sensors are emerging for tactile perception. Compared to traditional sensors entirely based on electronic materials[21,22], ionotronic sensors feature extraordinary softness, flexibility/stretchability, and optical transparency. Among the various types of ionotronic sensors, the capacitive type prevails owing to the ultrahigh sensitivity, high signal-to-noise ratio, low power consumption, and less sensitive to signal drifting[23]. An ionotronic capacitive sensor, similar to an electronic one, generally consists of different materials such as ionic conductors and dielectric elastomers. These constituents are often mechanically, electrically, and chemically dissimilar, yet their integration is indispensable during the fabrication of the sensor[1]. Such a process thus far has been primarily achieved via manual assembly of cast segments, which renders weak interfaces between constituents and restricts the design flexibility of the structures (mostly limited to planar architectures)[8,24] as well as the functionalities of the devices. Further development of the field requires advanced manufacturing techniques that integrate dissimilar materials with intricate architectures and strong adhesion. Three-dimensional (3D) printing technique readily meets these requirements by creating highly complex 3D objects in a layer-by-layer manner[25]. For example, direct ink writing (DIW)-based 3D printing creates 3D structures by extruding viscous ink through a nozzle, and has been used to manufacture heterogeneous gel-elastomer hybrids for stretchable ionotronic devices and soft robots[26–28]. Different from DIW-based 3D printing, digital light processing (DLP)-based 3D printing forms 3D structures by projecting digitalized ultraviolet (UV) patterns onto the surface of polymer resin to trigger localized photopolymerization, which converts the liquid resin into solid 3D structure[29]. Owing to its excellent capability of 3D printing structures with complex geometry and high resolution, DLP-based 3D printing has been extensively exploited in the recent past to manufacture various ionotronic sensors[30–35]. Nonetheless, current printed ionotronic sensors mainly consist of one single material, i.e. the ionic conductor, due to the lack of multi-material 3D printing capability that can seamlessly integrate ionic conductor and dielectric elastomer into one printed sensor. Moreover, the perceivable stimuli of these printed ionotronic sensors are mostly limited to compression and/or tension (Table S1). The deficiency of sensing capabilities greatly hampers the applications of ionotronic sensors in engineering, e.g. soft robots and human-machine interactions, where the sensors are desired to have more sophisticated capability to sense different mechanical stimuli such as tension, compression, shear, torsion, or even their combinations. Therefore, making reliable ionotronic capacitive sensors with multi-mode sensing capabilities is of significant importance for the field but remains an unresolved challenge.

Herein, using a printable polyelectrolyte elastomer and the DLP-based multi-material 3D printing technique, we resolve the challenge by fabricating ionotronic capacitive sensors with multi-mode sensing capabilities. We rationalize the structure-performance relationships and design a variety of architected sensors, achieving the sensing of tension, compression, shear, and torsion, with on-demand tailorable sensitivities. Moreover, we for the first-time design and manufacture integrated ionotronic sensors that can perceive different mechanical stimuli without mutual signal interferences, including the integrated tensile-compressive sensor, the integrated compressive-shear sensor,

and the integrated torsional-compressive sensor. We synthesize a photo-polymerizable ionic monomer for the polyelectrolyte elastomer, which is stretchable, transparent, ionically conductive, thermally stable, and leakage-resistant. We validate the printability of the polyelectrolyte elastomer using DLP-based 3D printing and show that the printed sensors possess robust interfaces between polyelectrolyte and dielectric elastomers. Furthermore, our experimental results indicate that the ionotronic capacitive sensors employing polyelectrolyte elastomers have extraordinary long-term stability owing to the free of leakage, while the traditional ionotronic capacitive sensors employing LiTFSI are prone to ion leakage and have poor stability. We design and print a sensor kit integrating four shear sensors and one compressive sensor, and demonstrate its applications as a remote-control unit for controlling the flight of a drone. The multi-material 3D printing of polyelectrolyte elastomers resolves the deficiencies of stability and functionalities simultaneously, opening ample space for the design and manufacturing of stretchable ionotronics for applications across a multitude of fields.

## Results and discussions

The high flexibility in structural design of multi-material 3D printing enables the fabrication of ionotronic sensors that mimic the sensing performances of the human skin. As shown in Fig. 1a, the human skin, which adopts ions as the charge carriers, contains various mechanoreceptors and is capable of multi-mode sensing, such as compression, tension, combined compression and shear, and combined torsion and compression (Fig. 1b). Notably, the human skin can perceive combined deformations meanwhile decipher individual stimuli without signal cross-talks. Mimicking the sensing capabilities of the human skin, we rationally design and print various architected ionotronic sensors with multi-mode sensing capabilities, including tension, compression, shear, torsion, combined tension and compression, combined compression and shear, and combined torsion and compression (Fig. 1c). To do so, we print the ionotronic sensors using a self-built DLP-based multi-material 3D printing system[36] and photo-curable precursors of polyelectrolyte elastomer (abbreviated as PEE hereafter) and dielectric elastomer (abbreviated as DE hereafter). The system employs the "bottom-up" projection approach where digitalized UV patterns are irradiated from the UV light engine, which is located below the printing stage that moves vertically to control the thickness of each slice. A glass plate holding two polymer precursor containers and moving horizontally to deliver the precursor is located between the UV light engine and the printing stage. An ionotronic capacitive sensor typically includes a layer of DE sandwiched by two layers of PEE. The PEE contains fixed anions (or cations) and mobile counterions, and is resistant to ion leakage. A robust interface between PEE and DE is generated owing to the covalent and topological interlinks formed during the printing. In general, ionotronics sensors are softer and more stretchable than their electronic counterparts[37,38]. In addition, since ionotronic sensors also employ ions as the charge carrier, they potentially provide a more seamless interface with the biological systems.

We first synthesize the photo-curable PEE (Fig. S1). Whereas PEE can be achieved by various chemistries[17], here we synthesize a pair of cation and anion, 1-butyl-3-methylimidazolium 3-sulfopropyl acrylate (abbreviated as BS hereafter) (Fig. 2a), which appears to be a transparent liquid (Fig. 2b). The anion of BS contains an acrylate functional group for free radical photo-polymerization. The composition and the chemical structure of BS are validated by the [1]H-NMR spectrum (Fig. S2). Since ionic species are hygroscopic in general, homopolymerization of BS yields a PEE that is prone to the erosion of water. The variation in water content will cause fluctuations in the properties of the PEE. To assuage this issue, we copolymerize BS with another hydrophobic acrylate-based monomer, ethylene glycol methyl ether acrylate (MEA), and use 1,6-hexanediol diacrylate (HDDA) as the crosslinker (Fig. 2c). After random copolymerization under UV

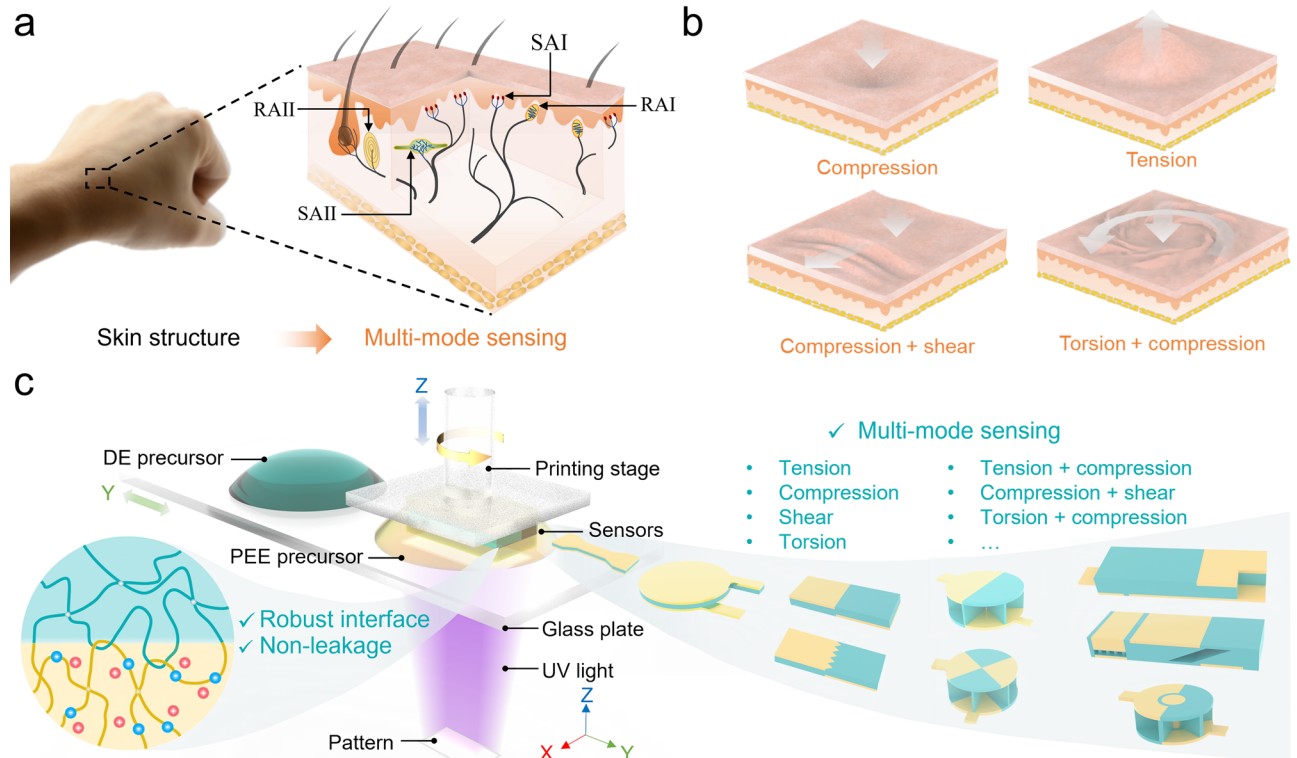

**Fig. 1 | Skin-mimicked ionotronic sensors with multi-mode sensing capabilities by DLP-based multi-material 3D printing. a** Schematic of the human skin containing various mechanoreceptors. The SAI responds to touch and static pressure, the SAII responds to stretching, the RAI responds to touch and dynamic pressure, and the RAII responds to deep pressure and vibration. **b** Human skin is capable of multi-mode sensing, such as compression, tension, combined compression and shear, and combined torsion and compression, without signal interferences. **c** 3D printing of various architected ionotronic sensors for multi-mode sensing using polyelectrolyte elastomers with robust interfaces and without leakage.

illumination (Fig. 2d), the poly(1-butyl-3-methylimidazolium 3-sulfopropyl acrylate-co-ethylene glycol methyl ether acrylate) (p(BS-co-MEA)) network is obtained with the anions engrafted to the backbone and the cations mobile (Fig. 2e). The weight loss of p(BS-co-MEA) due to the evaporation of water becomes lower than 5% when the molar ratio of BS:MEA is smaller than 1:1 (Fig. S3). FTIR spectra of the precursor of p(BS-co-MEA) before and after polymerization show that the absorption peak corresponding to the vinyl groups, ~1636 cm$^{-1}$, vanishes (Fig. 2f), indicating the complete conversion of the monomers.

The molar ratio between BS and MEA profoundly affects the mechanical and electrical properties of p(BS-co-MEA). Mechanically, we perform uniaxial tension for p(BS-co-MEA)s with various molar ratios of BS:MEA (Figs. 2g and S4). The fracture strain decreases while Young's modulus increases with BS:MEA (Fig. 2h). Electrically, the conductivity decreases with BS:MEA, since MEA is neutral, from $2.6 \times 10^{-2}$ S m$^{-1}$ to $3.4 \times 10^{-3}$ S m$^{-1}$ when the BS:MEA molar ratio is reduced from 1:0 to 1:10 (Fig. S5). The p(BS-co-MEA) samples are all highly transparent within the visible wavelength (Fig. 2i). For example, the transmittance of a 0.5-mm-thick sample at 600 nm is 91.3% for the molar ratio of BS:MEA = 1:1. After assessing the mechanical, electrical, and optical properties (Fig. S6), we optimally set the molar ratio of BS:MEA to be 1:1 and use the recipe to prepare the PEE precursor for the 3D printing of the ionotronic sensors hereafter. Thermogravimetric analysis reveals that the p(BS-co-MEA) with BS:MEA = 1:1 is thermally stable up to 275 °C (Fig. 2j), which should be sufficient for most engineering applications. We examine the solvent stability of PEE by sequentially soaking samples in water for 7 days, taking out and exsiccating the samples, and measuring the changes in mass and conductivity before soaking and after exsiccation. For comparison, we perform control experiments using a LiTFSI doped elastomer (abbreviated as LiE)[34]. It can be seen that the PEE maintains weight and conductivity well whereas the LiE dramatically loses weight and conductivity, by 26.6% and 98.4%, respectively (Fig. 2k). Soaking in an organic solvent, i.e. methyl cyanide (MeCN), leads to similar results that the weight and conductivity change negligibly for PEE but enormously for LiE (Fig. S7a). Besides, the PEE maintains shape well while the LiE becomes ragged after the test (Fig. S7b). Hence, it is noted that at the material level, the PEE is free of leakage, while the LiE is free of solvent leakage but prone to ion leakage.

We then explore the DLP-based multi-material 3D printing of ionotronic sensors using the PEE and a commercial acrylate elastomer, Tango (Stratasys Ltd., Eden Prairie, MN, USA) as the DE. Cytotoxicity tests indicate that both the PEE and the DE are biocompatible (Fig. S8). To make the precursors compatible with the printing system, 2, 4, 6-trimethylbenzoyl diphenylphosphine oxide (TPO) is used as the photo initiator[39]. We perform in-situ photo-rheological characterizations to investigate the photo-reactivity of PEE and DE. As shown in Fig. 3a, we identify the gelation time when the storage modulus curve intersects the loss modulus curve. To cure a 50 μm thick layer, the gelation time of DE is ~2 s and the gelation time of PEE is ~11 s, indicating that both materials are highly photocurable. Moreover, we carry out photo-rheological characterizations (Fig. S9) to determine the required gelation time (or energy density) to cure PEE or DE samples with different layer thicknesses. As shown in Fig. 3b, curing a thicker layer needs a longer gelation time. Specifically, owing to the high photo-sensitivity, the curing times for 300 μm thick samples are 45.8 s for PEE (under the irradiation of 405 nm UV projection at 384.7 mJ cm$^{-2}$) and 20.6 s for DE (under the irradiation of 405 nm UV projection at 173.0 mJ cm$^{-2}$). In addition, we further test the dual-material printability between PEE and DE by printing a grid pattern board where the width of the transparent DE line is 100 μm and the light-yellow PEE blocks are embedded in the

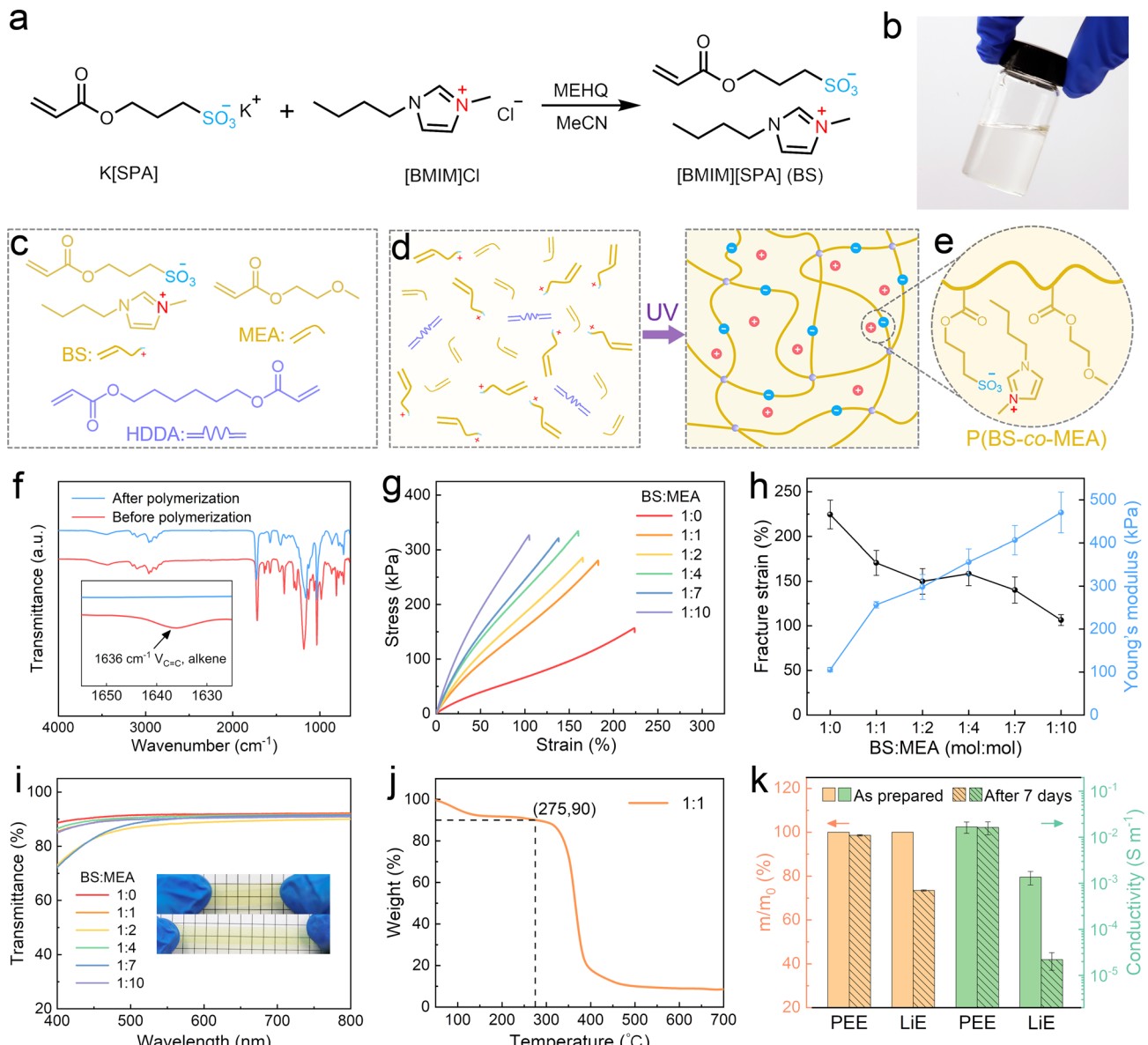

**Fig. 2 | Synthesis and characterizations of PEE. a** The synthesis of monomer, BS. **b** An image of the as-prepared BS. **c** Chemical structures of monomers, BS and MEA, and crosslinker, HDDA. **d** Synthesis of p(BS-co-MEA) network via photo-polymerization. **e** The p(BS-co-MEA) network contains engrafted negative charges and mobile positive charges. **f** FTIR spectra of the precursor of p(BS-co-MEA) before and after polymerization. The inset shows the disappearance of the peak corresponding to the vinyl group. **g** Nominal uniaxial tensile stress-strain curves of p(BS-co-MEA) with various molar ratios of BS:MEA. **h** The variations of fracture strain and Young's modulus with the molar ratios of BS:MEA. The error bars represent standard deviations. **i** Transmittance of various p(BS-co-MEA). **j** TGA measurement of p(BS-co-MEA) with the molar ratio of BS:MEA = 1:1. **k** The changes in weight (orange) and conductivity (green) of PEE and LiE after soaking in water for 7 days and drying. The error bars represent standard deviations.

DE grids (Fig. 3c). The printed PEEs have comparable mechanical properties as those of the cast PEEs (Fig. S10a) and exhibit excellent elasticity and mechanical stability. Negligible hysteresis or degradation in modulus can be detected from the stress-strain curves after 300 cycles of cyclic stretch with a maximum strain of 100% (Fig. S10b). Uniaxial tensile tests give Young's moduli, obtained by linear fitting of the stress-strain curves in the strain range of 0-5%, of $156 \pm 19$ kPa, $799 \pm 57$ kPa, and $526 \pm 33$ kPa for PEE, DE, and sensor, respectively (Fig. S11). Provided that the thicknesses of PEE and DE are 0.2 mm and 0.4 mm, the weighted average modulus of the sensor is 588 kPa, in good agreement with the experiment.

Since the ionotronic capacitive sensors are multi-layer laminates, a robust interface between DE and PEE layers is important to avert interfacial delamination. Whereas conventional manufacturing techniques are struggling to create robust interfaces between dissimilar polymeric materials, strong interfacial bonding is readily achieved by multi-material 3D printing since each layer is only partially cured during one exposal of UV light and gets engaged in the polymerization of the subsequently printed layer. We perform the 180° peeling tests to assess the interfacial toughness of 3D printed and manually assembled PEE-DE bilayers (Fig. 3d). The adhesion energies, given by the plateaued normalized force, $2F_{ss}/W$, where $F_{ss}$ is the steady-state peel force and $W$ is the width of the sample (Fig. S12), are 339.3 J m$^{-2}$ and 4.1 J m$^{-2}$ for printed and assembled samples, respectively. Cohesive rupture occurs during the peeling of the printed bilayer that PEE residues are left on the surface of DE after peeling (Fig. 3e), meaning that the interface is tougher than the bulk of PEE[40]. The strong interfacial bonding is mainly ascribed to the topological entanglements due to the similar chemistries between PEE and DE and the covalent interlinks due to the partial curing of each printing layer. By contrast,

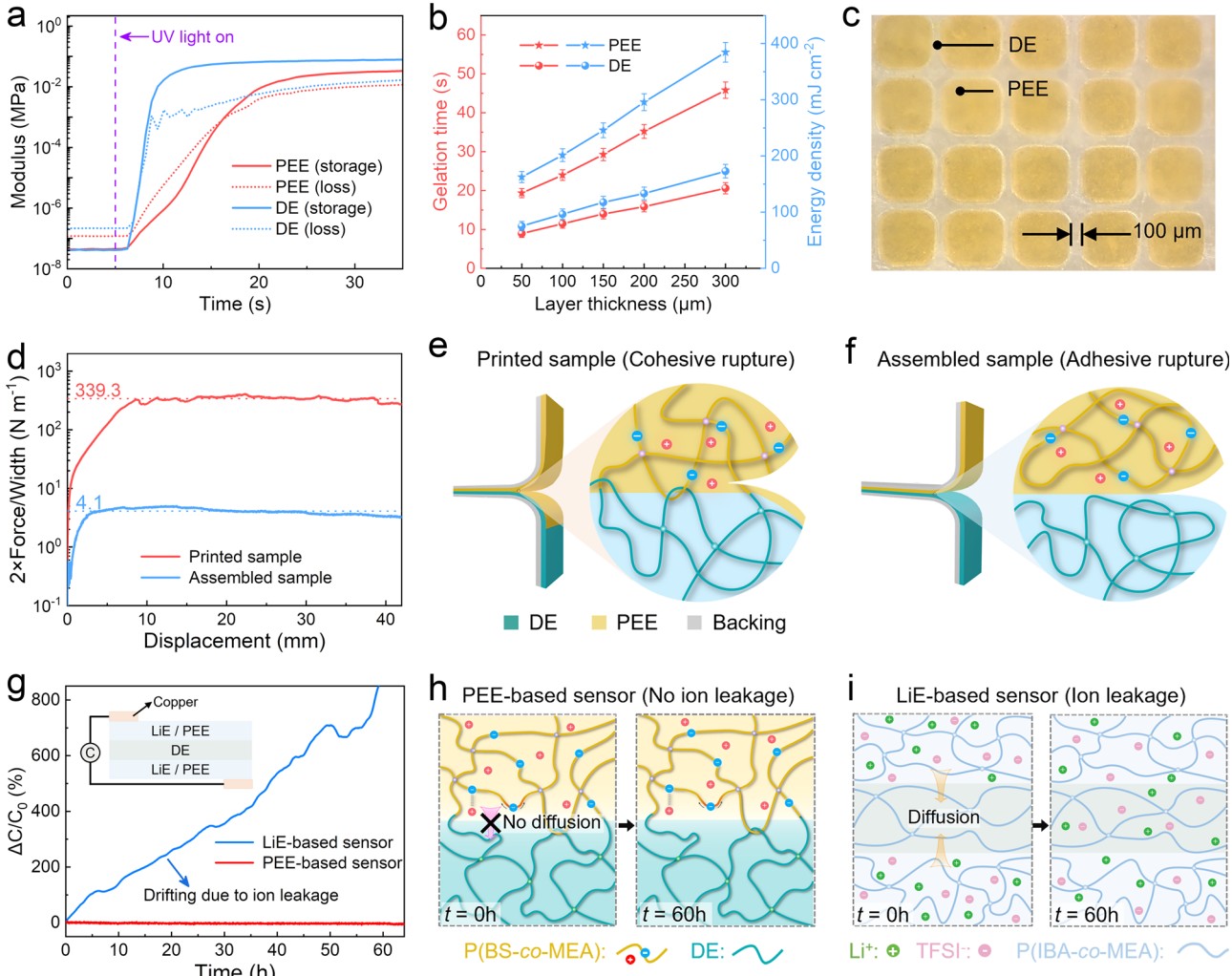

**Fig. 3 | Characterizations of printing and printed ionotronic sensors. a** Storage modulus and loss modulus of PEE and DE vary with time. **b** The variations of gelation time and energy density with layer thickness for PEE and DE. The error bars represent standard deviations. **c** Microscopic image of a PEE and DE grid pattern. The width of the grid is 100 μm and the length of the square is 500 μm. **d** 180° peeling curves for 3D-printed (red) and manually assembled (blue) PEE/DE bilayers. Schematics showing (**e**) the cohesive rupture of printed PEE/DE bilayer, and (**f**) the adhesive rupture of assembled PEE/DE bilayer. **g** Variation of $\Delta C/C_0$ with time for PEE-based sensor and LiE-based sensor. Schematics illustrating (**h**) the prohibition of ion leakage in PEE-based sensor, and (**i**) the ion leakage in LiE-based sensor.

the manually assembled bilayer experiences adhesive rupture with low adhesion energy (Fig. 3f).

Recall that we have examined the stability at the material level for PEE and LiE. Now we probe the stability at the device level. We fabricate an ionotronic sensor by sandwiching a layer of DE with two layers of LiE or PEE and measure the capacitance over time. As shown in Fig. 3g, the normalized change in the capacitance, $\Delta C/C_0$ ($\Delta C$ is the change of capacitance and $C_0$ is the original capacitance), of the PEE-based sensor is stable over time, while the $\Delta C/C_0$ of the LiE-based sensor keeps drifting until short-circuit sets in. Further monitoring reveals that the PEE-based sensor maintains stability over ~11 months in the ambient environment (Fig. S13). The results are interpreted as follows. When in contact with DE, the ions in the PEE tend to diffuse toward the DE due to the concentration gradient. However, the directional diffusion of the fixed anions exerts tensile stress on the polymer chains of PEE, which counteracts the chemical potential of the anions to prevent long-range diffusion (Fig. 3h). Meanwhile, the directional diffusion of the mobile cations is also prohibited due to the electrostatic interactions. Consequently, PEE is resistant to ion leakage and the PEE-based ionotronic sensor is stable. By contrast, both Li⁺ and TFSI⁻ ions are mobile and diffuse into the DE layer (Fig. 3i). After short circuit, the DE layer becomes conductive and the external circuit measures the

resultant capacitance of the two serial capacitors of electric double layer, which is formed at the interface between an electronic conductor and an ionic conductor[8]. We further verify this hypothesis by separately synthesizing a piece of LiE with the same ion concentration, connecting it to two metal electrodes with the same area of electric double layer, and measuring the capacitance to be ~0.94 nF, which is comparable to the plateaued capacitance of the LiE-based sensor, ~1 nF (Fig. S14). The diffusion coefficient of LiTFSI in DE can be estimated by $L^2/\tau = 7.23 \times 10^{-13}$ m² s⁻¹, where $L$ is the thickness of the DE layer and $\tau$ is the diffusion time, which is consistent with literature[41]. Hence, it is noted that at the device level, the PEE-based sensor is stable while the LiE-based sensor suffers signal drift due to ion leakage.

First-generation stretchable ionotronic sensors have been explored to mimic neurosensory systems. However, previously reported sensors rarely sense mechanical stimuli except for tension and compression[1,7]. The paucity of sensing diversity greatly hinders the close imitation of neurosensory systems such as the human skin, which can perceive not only tension and compression, but also shear, torsion, and the combination thereof. The DLP-based multi-material 3D printing enables the seamless integration of PEE and DE into a single ionotronic sensor and allows the design and fabrication of ionotronic sensors with multi-mode sensing capabilities. Fig. 4a schematizes a

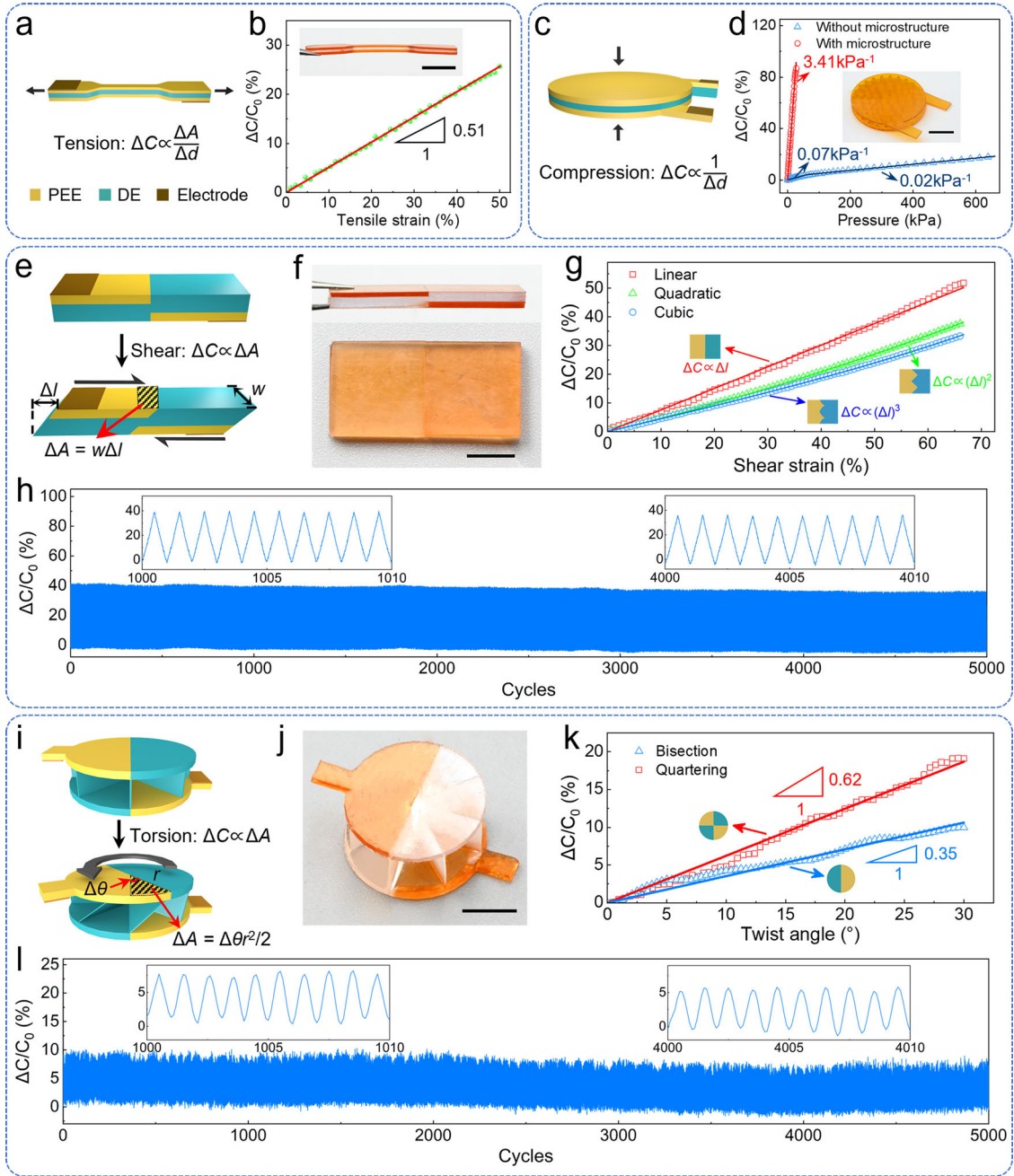

**Fig. 4 | Ionotronic sensors for tension, compression, shear, and torsion.**
**a** Structure and principle of the tensile sensor. **b** Variation of $\Delta C/C_0$ with tensile strain. The inset shows the snapshot image of a tensile sensor. **c** Structure and principle of the compressive sensor. **d** Variations of $\Delta C/C_0$ with pressure for the compressive sensors with/without microstructures. The inset shows the snapshot image of a micro-structured sensor with a diameter of 20 mm. **e** Structure and principle of the shear sensor. **f** Snapshot images of a shear sensor. **g** Variation of

$\Delta C/C_0$ with shear strain for the shear sensors with different profiles as indicated. **h** Cyclic shear test with a maximum shear strain of 66.7%. **i** Structure and principle of the torsional sensor. **j** Snapshot image of a torsional sensor. **k** Variation of $\Delta C/C_0$ with the angle of twist for the torsional sensors with different profiles as indicated. **l** Cyclic torsional test of the bisected torsional sensor with a maximum twist angle of 30°.

dumbbell-shaped tensile sensor. Subject to uniaxial tension, the area increases and the thickness shrinks such that the capacitance increases, $\triangle C \propto \triangle A/\triangle d$. The sensor exhibits excellent linear response up to 50% tensile strain with a sensitivity (defined as $\delta(\Delta C/C_0)/\delta\varepsilon$, where $\varepsilon$ is the tensile strain) of 0.51 (Fig. 4b). Finite element analysis (FEA) reveals that, whereas the majority of deformation occurs in the central segment, $\Delta C/C_0$ varies with tensile strain mostly in a linear manner, which is in satisfactory agreement with the experiment (Fig. S15). Subject to cyclic tension with a maximum strain of 30%, the sensor maintains mechanical and electrical stability, showing negligible degradation of

stress and a tiny drift of capacitance within 2% over 1800 cycles (Fig. S16). Fig. 4c schematizes a circular compressive sensor. A compressive force reduces the thickness of DE layer and changes the area negligibly, and the capacitance of the sensor rises accordingly, $\triangle C \propto 1/\triangle d$. Whereas the sensor with homogeneous DE layer has poor sensitivity due to the high stiffness and incompressibility of DE, 3D printing provides a facile manner to endow the DE layer with microstructures, reducing the stiffness of the DE layer and thus improving the sensitivity of the sensor. Fig. 4d plots the variations of $\Delta C/C_0$ with pressure (*P*) for the sensors with/without microstructures. The micro-

structured sensor has a sensitivity of 3.41 kPa$^{-1}$, higher than that of the sensor without microstructure by two orders of magnitude. Again, the prominent durability of the sensor is manifested by the exceptional stability over 10000 cycles of cyclic loading (Fig. S17).

3D printing enables the fabrication of ionotronic sensors with unusual configurations to detect shear and torsion. Fig. 4e schematizes a shear sensor. Subject to a shear force, the overlap area of PEEs increases and the capacitance of the sensor increases, $\triangle C \propto \triangle A$. Fig. 4f displays the side and top views of a printed shear sensor. For a differential increment in shear displacement d$l$, the overlap area increases by

$$\triangle A = \int w(l)dl \qquad (1)$$

where $w$ is the characteristic dimension in the width direction. For a flat front line, $w(l)$ is a constant (zero order), so the overlap area increases linearly with the shear displacement, which in turn is approximately linearly proportional to the shear strain at small deformation. Consequently, $\Delta C/C_0$ varies mostly linearly with the shear strain $\gamma$, as validated by the red data in Fig. 4g. Notably, the sensitivity (defined as $\delta(\Delta C/C_0)/\delta\gamma$) of the shear sensor can be tailored on demand by programming the pattern of the front line, owing to the high design and manufacturing flexibility enabled by the multi-material 3D printing. As shown in Fig. S18, when the front line is in a zigzag pattern, i.e. $w(l)$ varies linearly with $l$ (first order), $\Delta C/C_0$ will vary quadratically with the shear strain (green data in Fig. 4g); when the front line is in a parabolic pattern, i.e. $w(l)$ varies quadratically with $l$ (second order), $\Delta C/C_0$ will vary cubically with the shear strain (blue data in Fig. 4g). In principle, any sensitivity can be realized so long as the feature size of the pattern of the front line is within the resolution of the printing system. Subject to a cyclic shear test with a maximum shear strain of 66.7%, both the electrical responses (Fig. 4h) and mechanical responses (Fig. S19) of the shear sensor remain stable up to 5000 cycles.

Fig. 4i schematizes a torsional sensor. Subject to a torsion, the overlap area of the bisected PEEs increases and the capacitance of the sensor increases, $\triangle C \propto \triangle A$. Fig. 4j shows the snapshot image of a printed torsional sensor. The multi-material 3D printing not only allows making two distinct materials in the same layer with seamless interfaces, which is extremely challenging for other manufacturing techniques, but also readily engenders pliable architected DE layer and strong interfaces. Furthermore, the sensitivity of the torsional sensor (defined as $\delta(\Delta C/C_0)/\delta\theta$, where $\theta$ is the angle of twist) can be tuned on demand through exquisite structure design as well (Fig. S20). With an increment in twist angle, $\Delta\theta$, the overlap area increases by

$$\triangle A = n\triangle\theta r^2/4 \qquad (2)$$

where $n$ is the number of divisions and $r$ is the radius of the sensor. Since the capacitance is positively related to the overlap area, the sensitivity of the sensor is also positively related to the number of divisions. As shown in Fig. 4k, the sensor has a sensitivity of 0.35 for a bisection pattern and 0.62 for a quartering pattern. Subject to a cyclic torsional test with a maximum twist angle of 30°, the sensor with a bisection pattern maintains excellent stability for 5000 cycles (Fig. 4l).

The human skin is multifunctional to perceive various mechanical perturbations both individually and simultaneously, yet such multi-mode sensing capability has not been achieved in ionotronic sensors. Although a tensile sensor may also be used to sense compression, the signals from tension and compression confound each other and are difficult to decouple upon combined loadings. Here we demonstrate integrated ionotronic sensors with multi-mode sensing capability yet without mutual signal interferences. We first design and fabricate an integrated ionotronic sensor that can decipher the signals of compression, tension, or their combination. The design and principle of

the sensor are sketched in Fig. 5a. The right part constitutes a compressive sensor monitored by the capacitance meter $C_1$, and the bottom part constitutes a tensile sensor monitored by the capacitance meter $C_2$. Note that the two sensors have one shared electrode and two independent electrodes, which are separated much farther than the thickness of the DE layer to minimize signal cross-talks. In addition, the DE layer of the tensile sensor is much thinner than the DE layer of the compressive sensor such that, the relative thickness change of the DE layer of the compressive sensor is negligible when the tensile sensor is activated. Fig. 5b shows the snapshot image of a printed integrated tensile and compressive sensor. Fig. 5c presents the equivalent circuit diagram of the sensor where the PEEs are treated as pure resistors ($R_1$, $R_2$, and $R_3$), the DEs are treated as adjustable capacitors ($C_1'$ for the compressive unit and $C_2'$ for the tensile unit), and the electric double layers are treated as pure capacitors ($C_{EDL}$). Capacitor $C_1'$ and capacitance meter $C_1$ form the subcircuit of the compressive sensor. Capacitor $C_2'$ and capacitance meter $C_2$ form the subcircuit of the tensile sensor. Since the shared elements, $R_2$ and $C_{EDL}$, are constant, the two subcircuits are independent of each other. We apply single loading to probe the responses of the sensor (Fig. S21). As shown in Fig. 5d, under 25% compressive strain, $C_1$ increases by 27.87% and $C_2$ increases by 2.22%, giving a signal ratio of 12.6; under 50% tensile strain, $C_2$ decreases by 10.14% while $C_1$ increases by 0.32%, giving a signal ratio of 31.7. The associated ghost change of capacitance should result from the Poisson's effect. Nevertheless, the signal ratios in the two single loading modes are sufficiently large for decoupling. Besides, the opposite tendency of capacitance change under tension is also beneficial for signal decoupling. FEA results also show prominent differences between the signals of $C_1$ and $C_2$ when the sensor is subject to compression (Fig. S22) or tension (Fig. S23). The key point of signal decoupling is to minimize the associated deformation of one sensor when another sensor is deformed through appropriate structure design. As a counterexample, both the capacitances of $C_1$ and $C_2$ of a badly designed sensor change notably with tensile strain (Fig. S24). We carry out 10 cycles of loading and unloading with different strains and monitor the real-time capacitance change. As presented in Fig. 5e, the compression strains result in significant increases in $\Delta C/C_0$ for $C_1$ while the increases in $\Delta C/C_0$ for $C_2$ are lower by one order of magnitude. Similarly, apparent decreases in $\Delta C/C_0$ for $C_2$ can be observed, but the signal variation for $C_1$ is negligible when the sensor is under tension (Fig. 5f). The outstanding repeatability of the signals further justifies the stability of the sensors. To demonstrate the sensor's excellent capability of decoupling the signals from combined deformation, we apply a series of combined compression (ranging from 0% to 33%) and tension (ranging from 0% to 100%) to the sensor. It can be seen that only compression could lead to a notable change in $\Delta C/C_0$ for $C_1$ and only tension could induce a notable decrease in $\Delta C/C_0$ for $C_2$ (Fig. 5g). Therefore, any combination of compression and tension can be decoded by comparing the two plots in Fig. 5g.

The versatility of multi-material 3D printing allows us to design and fabricate other integrated ionotronic sensors. We demonstrate two more types of integrated ionotronic sensors, one can decipher the signals of compression, shear, or their combination, and the other can decipher the signals of torsion, compression, or their combination (Fig. S25). Fig. 5h schematizes the design and principle of the integrated compressive and shear sensor. The sensor senses compression through the capacitance meter $C_3$ and shear through the capacitance meter $C_4$. The DE layers of both sensing units are designed to be hollow to improve compliance. The DE layer of the compressive unit is made micro-structured to further improve compliance. Furthermore, it should be noted that the thickness of the DE layer of the compressive unit should be much thinner than that of the shear unit (Fig. S25a). Assume that both sensing units can be equivalent to a plane-parallel capacitor, with the thicknesses of the DE layer being $d_3$ and $d_4$ ($d_3 \ll d_4$), respectively. Subject to compression, the overall sensor is flattened by

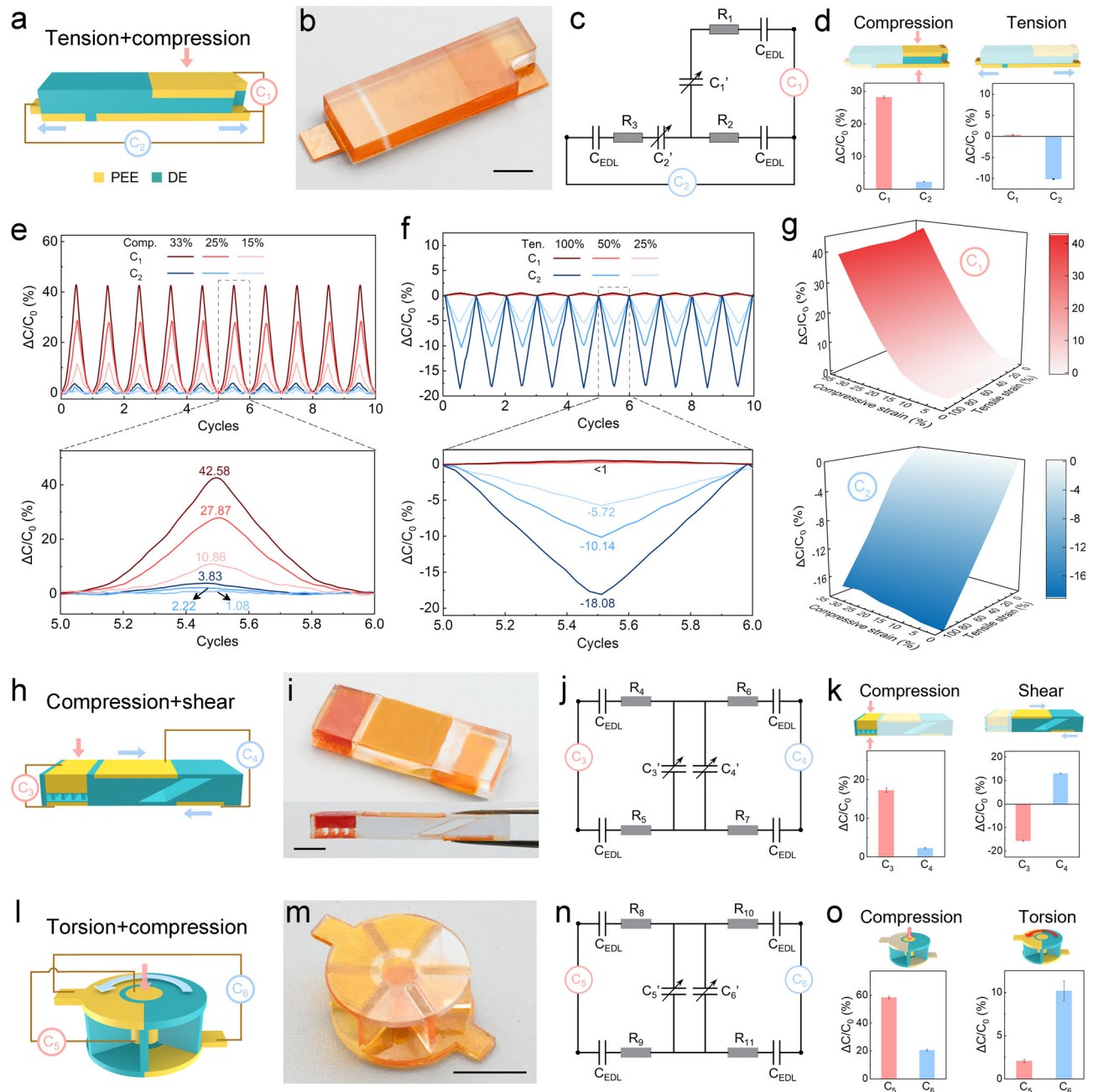

**Fig. 5 | Various integrated ionotronic sensors. a** Design and principle of the integrated tensile and compressive sensor. Capacitance meter $C_1$ measures compression and capacitance meter $C_2$ measures tension. **b** Snapshot image of a printed sensor. **c** Equivalent circuit diagram of the sensor. **d** $\Delta C/C_0$ of $C_1$ and $C_2$ when the compressive unit or the tensile unit is activated. The error bars represent standard deviations. Responses of the sensor subjected to 10 cycles of (**e**) compression, and (**f**) tension with various strains as indicated. **g** The signal maps of $C_1$ and $C_2$ under the combined deformation of compression and tension. **h** Design and principle of the integrated compressive and shear sensor. Capacitance meter $C_3$

measures compression and capacitance meter $C_4$ measures shear. **i** Snapshot images of a printed sensor. (**j**) Equivalent circuit diagram of the sensor. **k** $\Delta C/C_0$ of $C_3$ and $C_4$ when the compressive unit or the shear unit is activated. **l** Design and principle of the integrated torsional and compressive sensor. Capacitance meter $C_5$ measures compression and capacitance meter $C_6$ measures torsion. **m** Snapshot image of a printed sensor. **n** Equivalent circuit diagram of the sensor. **o** $\Delta C/C_0$ of $C_5$ and $C_6$ when the compressive unit or the torsional unit is activated. The error bars represent standard deviations.

$\Delta d$. When the relative thickness change of the compressive unit ($\Delta d/d_3$) is large enough to induce an appreciable signal in the compressive unit, the signal in the shear unit is much smaller since $\Delta d/d_4 \ll \Delta d/d_3$. Fig. 5i shows the snapshot images of a printed integrated compressive and shear sensor. Fig. 5j presents the equivalent circuit diagram of the sensor, with $C_3'$ representing the DE layer of the compressive unit and $C_4'$ representing the DE layer of the shear unit. As shown in Fig. 5k, when the sensor is compressed by 12.5%, $C_3$ increases by 17.25% and $C_4$ increases by 2.29%, giving a signal ratio of 7.53; when the sensor is

sheared by 50%, $C_4$ increases by 13.04% while $C_3$ decreases by 15.63%. Whereas the magnitudes of the signals of the two sensing units are comparable under shear, signal decoupling is feasible because of the opposite tendencies in capacitance change. Therefore, the signals caused by compression and shear are decoupled (Movie S1). Fig. 5l schematizes the design and principle of the integrated torsional and compressive sensor. The sensor senses compression through the capacitance meter $C_5$ and torsion through the capacitance meter $C_6$. Similarly, the DE layers of both sensing units are designed to be hollow

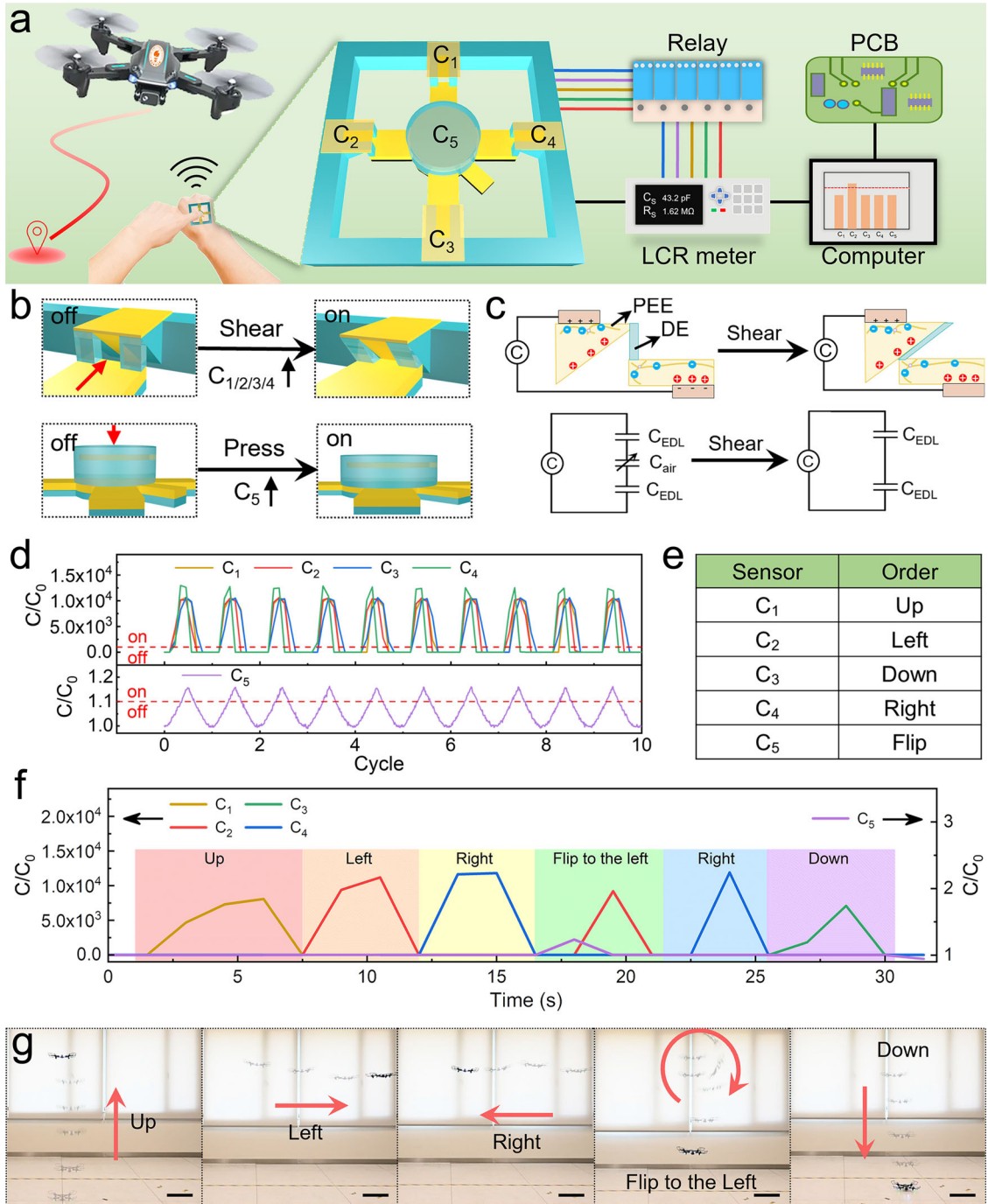

**Fig. 6 | Printed wearable wireless remote-control unit for a drone. a** The block diagram of a controller system for a drone that can be controlled by a printed wearable wireless remote-control unit, which contains five ionotronic sensors as indicated. **b** Schematics of the operation modes of the shear sensors ($C_1$, $C_2$, $C_3$, and $C_4$) and the compressive sensor ($C_5$). **c** Working mechanism of the shear sensor. **d** $\Delta C/C_0$ varies with testing cycle. The red dash line indicates the threshold, above which the corresponding sensor is switched on. (**e**) The coding of the remote-control unit. **f** Real-time responses of five sensors during a trial. **g** Sequential snapshots showing the reactions of a drone to the commands of the remote-control unit.

to improve compliance. Also, the thickness of the DE layer of the compressive unit is purposely designed to be much thinner than that of the torsional unit (Fig. S25b) such that, compression will induce an appreciable signal in the compressive unit but not in the torsional unit. Fig. 5m shows the snapshot image of a printed integrated torsional and compressive sensor. Fig. 5n presents the equivalent circuit diagram of the sensor, with $C_5'$ representing the DE layer of the compressive unit and $C_6'$ representing the DE layer of the torsional unit. As shown in Fig. 5o, when the sensor is compressed by 40%, $C_5$ increases by 58.3% and $C_6$ increases by 20.69%, giving a signal ratio of 2.82; when the

sensor is twisted by 60º, $C_5$ increases by 2.07% and $C_6$ increases by 10.18%, giving a signal ratio of 4.92. The signals caused by compression and torsion are decoupled as well.

By virtue of their exceptional mechanical and electrical compliance to tissues, stretchable ionotronic devices are promising candidates to interface with the human body. As a proof of concept, we demonstrate a wearable remote-control unit that can wirelessly communicate with a drone. The block diagram of the controller system is outlined in Fig. 6a. The remote-control unit integrates five sensors: one compressive sensor and four shear sensors (Fig. S26), which are used

as the input ports. Upon activation, each sensor generates a signal of capacitive change, which is fed through a time relay and then to an LCR meter. A customized LabVIEW controlling program collects and processes the signal and sends it to a printed circuit board (PCB), which further generates a command to the drone via electromagnetic waves (Fig. S27). Fig. 6b delineates the operation modes of the shear sensor and the compressive sensor. Unlike the one containing a capacitor due to the DE layer in Fig. 4e, the shear sensor of the remote-control unit accommodates a capacitor due to the air in series with two capacitors due to the electric double layer (Fig. 6c). Upon shear, the PEEs come into contact with each other to eliminate the air capacitor, resulting in a giant capacitance change by orders of magnitude. The compressive sensor, similar to that of Fig. 4c, consists of two layers of PEEs sandwiching a layer of DE. Its capacitance increases in response to applied pressure. We characterize the responses of the five sensors (Movie S2) and plot the variations of their normalized capacitance over 10 cycles of loading and unloading (Fig. 6d). The capacitance of the shear sensors after the switch-on increases by more than $10^4$ times. Such a giant signal is noise tolerant and highly beneficial for circuit design. We set the threshold for switching on to be 1000 for the shear sensors and 1.1 for the compressive sensor. It should be noted that the threshold of 1.1 for the capacitive sensor is mainly limited by the intact DE layer yet is still adequate for programming. A higher threshold can be achieved by architecting microstructures in the DE layer of the capacitive sensor (Fig. 4d). The mapping relationships between the sensors of the remote-control unit and the orders for the drone are listed in Fig. 6e. We sequentially activate the sensors of $C_1$, $C_2$, $C_4$, $C_5 + C_2$, $C_4$, and $C_3$, corresponding to the orders of "up", "left", "right", "flip to the left", "right", and "down". The real-time signals are displayed in Fig. 6f and the snap-shots of the flight of the drone are captured in Fig. 6g. Excellent maneuverability of the drone in response to the steering signals of the remote-control system is demonstrated (Movie S3). Furthermore, the adhesion between the remote-control unit and the substrate is important in practical deployments. Similar to the robust adhesion achieved between the PEE and DE in the printed sensors, achieving robust adhesion between the sensor and other materials is feasible using the DLP-based 3D printing. As an example, we print a layer of polyacrylamide hydrogel to strongly adhere a printed sensor to various materials, including fabric, skin, plastic, and metal (Fig. S28).

In summary, we report a variety of ionotronic sensors capable of multi-mode sensing fabricated by DLP-based multi-material 3D printing using a printable polyelectrolyte elastomer. The polyelectrolyte elastomer is free of leakage, contributing to the extraordinary long-term stability of the sensors. The multi-material 3D printing capability allows high flexibility in the structural design of the sensors, enabling the sensing of different mechanical stimuli, both individually and their combinations without signal crosstalk, and tailorable sensitivities through the elaborate programming of the device architectures. A wearable remote-control unit integrating five ionotronic sensors for a drone is demonstrated, signifying the enormous potential of stretchable ionotronics based on the multi-material 3D printing of polyelectrolyte elastomers for assorted applications across fields.

## Methods

### Materials
Ethylene glycol methyl ether acrylate (MEA), 1-butyl-3-methylimidazolium chloride ([BMIM]Cl), isobornyl acrylate (IBA), polyethylene (glycol) diacrylate (PEGDA, molecular weight 200), and photo-initiator 2,4,6-trimethylbenzoyl diphenylphosphine oxide (TPO) were purchased from Sigma-Aldrich, 3-sulfopropyl acrylate potassium salt (K[SPA]), methoxyphenol (MEHQ), and 1,6-hexanediol diacrylate (HDDA) were purchased from Shanghai Aladdin, and photo-initiator benzophenone (BP) was purchased from Shanghai Macklin. All chemicals were used as received without further purification.

### Synthesis of BS
To synthesize BS, 8.73 g [BMIM]Cl (50.0 mmol), 11.6 g K[SPA] (50.0 mmol) and 11.6 mg MEHQ as an inhibitor were dissolved in 35 mL of acetonitrile. The mixture was stirred for 24 hours at room temperature, after which the precipitated potassium chloride (KCl) was filtered. Then the solution was concentrated under reduced pressure to obtain a clear viscous liquid of BS (15.4 g, yield 93%). The composition and chemical structure are confirmed by the nuclear magnetic resonance (NMR, NMR-400M (AVANCE III 400 M), SUSTech Core Research Facilities), and the $^1$H NMR (D$_2$O, 400 MHz) spectra are: 8.75 (s, 1H), 7.49 (d, J = 19.1 Hz, 2H), 6.47 (d, J = 17.3 Hz, 1H), 6.24 (dd, J = 17.3, 10.5 Hz, 1H), 6.02 (d, J = 10.5 Hz, 1H), 4.32 (t, J = 6.3 Hz, 2H), 4.23 (t, J = 7.1 Hz, 2H), 3.92 (s, 3H), 3.08 − 3.00 (m, 2H), 2.22 − 2.11 (m, 2H), 1.92 − 1.83 (m, 2H), 1.42 − 1.28 (m, 2H), 0.95 (t, J = 7.4 Hz, 3H)

### Cast synthesis of PEEs
Monomers BS and MEA were mixed at a certain ratio (1:0, 1:1, 1:2, 1:4, 1:7, 1:10). Then 0.5 mol% HDDA as the crosslinker and 0.5 mol% photo-initiator, with respect to the total amount of the monomer, were added to form a transparent precursor solution. The solution was injected in a mold made by sandwiching a 0.5 mm thick poly(tetrafluoroethylene) (PTFE) spacer with two glass plates and subjected to UV light irradiation with a wavelength of 365 nm for 2 hours.

### Characterizations of PEEs
For mechanical characterizations, the PEEs were cut into dumbbell shape with a gauge length of 12 mm, width of 2 mm, and thickness of 0.5 mm. The samples were loaded to a universal testing machine (Instron 5966) with a 100 N loading cell and pulled at a loading speed of 50 mm min$^{-1}$ up to rupture in monotonic tests. In cyclic tests, the loading speed was 6 mm s$^{-1}$. Unless otherwise specified, each experiment was repeated at least three times. For electrical characterizations, The PEEs were cut into a rectangular sample with dimensions of 40 mm × 5 mm × 0.5 mm and their resistances were measured by a bench multimeter Keysight E4990A using the four-point probe method. The conductivity was calculated by $\sigma = \frac{L}{AR}$, where $L$ denotes the distance between the electrodes, $A$ denotes the cross-sectional area of the sample, and $R$ is the resistance measured by the multimeter.

### FTIR spectroscopy
The FTIR Spectra were recorded by using an FTIR spectrophotometer (Thermo Nicolet iS50), ranging from 400 to 4000 cm$^{-1}$.

### Transmittance measurement
The PEE samples with different molar ratios of BS:MEA were cut into 30 mm × 15 mm × 0.5 mm. The samples were loaded to the UV−visible spectrophotometer (Metash SH, UV8000) and scanned from 400 to 800 nm at room temperature.

### TGA Measurement
The TGA was measured by using TG 209F1 with a scanning rate of 10 °C min$^{-1}$ from 50 to 700 °C under N$_2$ atmosphere.

### Cytotoxicity test
To test the cell viability of PEE and DE, we cultured the NIH 3T3 cells (Mouse embryonic fibroblast) with the PEE and DE for a period of time. NIH 3T3 cells were cultured in Dulbecco's modified eagle medium (DMEM, high glucose) supplemented with 10% fetal bovine serum (FBS) and 1% penicillin-streptomycin at 37 °C in a humidified chamber containing 5% CO$_2$. PEE and DE slabs with dimensions of ~ 5 mm in length, ~ 5 mm in width, and 1 mm in thickness were immersed in the cell culture medium in a 48-well plate for 24 hours. Then the NIH 3T3 cells were treated with PEE and DE for 24 hours. The Calcein AM Cell Viability Assay Kit (stain living cells) was used to test the viability of cells incubated with PEE and DE for 24 hours.

## Solvent stability

To synthesize LiE, LiTFSI was added into the solvent of MEA and IBOA (molar ratio of MEA:IBOA = 4:1) at a concentration of 1.5 M. After adding the photoinitiator 1173 (0.5 mol% in respect to MEA), the mixture was exposed to ultraviolet light irradiation (365 nm, 15 W) for 12 hours. LiE and PEE samples (30 mm × 10 mm × 0.5 mm) were immersed in 50 mL water and acetonitrile for 7 days and then transferred to a 65 °C oven overnight for complete drying. The conductivity and mass of the samples before and after the immersion were measured.

## Weight loss of PEEs

The PEEs samples equilibrated in the open air (RH ~ 70%) were transferred to a 65 °C oven for 4 hours. Then the weight of the samples was measured and the weight loss was calculated as $\Delta m/m_0 = (m_0-m)/m_0 \times 100\%$, where $m_0$ is the initial weight and m is the weight after desiccation.

## 3D printing of sensors

We printed all types of sensors and the remote-control unit using a self-built DLP-based multi-material 3D printing system. Details about the printing system can be found in our previous articles[42]. In brief, the system consists of a light engine (PRO4710, Wintech, San Marcos, CA, USA) and two translational stages (LTS150, Throlabs Inc., Newton, NJ, USA). The vertical translational stage controls the position of the printing platform and the thickness of each printed layer. The horizontal translational stage connects a glass plate that supports two polymer precursor containers, and moves horizontally to deliver a needed polymer precursor for the corresponding layer. The top surface of the glass plate is covered by polytetrafluoroethylene (PTFE) tape to minimize the adhesion between the printed structure and the glass plate. The 3D printing system is controlled in sequence using codes written in LabVIEW 2020 (National Instruments, Austin, TX, USA). We optimized the printing area, optical resolution, and layer thickness to print the sensors and remote-control unit. For printing the sensors, the printing area was 34 mm × 19 mm, the optical resolution was 17.6 μm per pixel, and the layer thickness was 50 μm. For printing the remote-control unit, the printing area was 68 mm × 38 mm, the printing resolution was 35.2 μm per pixel, and the layer thickness was 100 μm. After printing, we used ethanol to rinse the printed structure and remove the uncured precursor. Subsequently, we placed the printed structure in a UV oven with 365 nm wavelength for 1 hour for complete curing of the elastomers.

## Rheological measurements

The in-situ photo-rheological properties of PEE and DE were measured using a rheometer (TA DHR3) with a parallel plate geometry. A UV light source (405 nm, 8.4 mW cm$^{-2}$) was attached to the rheometer to illuminate UV light during the curing.

## Curing thickness

The prepared PEE and DE precursors were respectively sandwiched between two glass slides with a gap ranging from 50 to 300 μm, and then the patterned near UV light (405 nm, 19.34 mW cm$^{-2}$) was irradiated. The curing time was recorded when a pattern could be visually observed.

## Peeling test

180° peel test was performed to measure the adhesion energy between PEE and DE, using the Instron 5966 with a 100 N load cell at a constant peeling speed of 30 mm min$^{-1}$. A stiff backing was bonded to each layer of samples using double-sided mesh tape. For the printed sample, the sample was printed with dimensions of 60 mm × 10 mm × 1 mm (0.5 mm for each layer) and a 10 mm long pre-crack was made. For the assembled sample, the sample was prepared by assembling a piece of PEE (60 mm × 10 mm × 0.5 mm) and a piece of DE (60 mm × 10 mm × 0.5 mm) and then the sample was stood for 30 minutes before the test.

## Characterizations of sensors

All sensors were designed, printed, and connected to the external circuit via copper wires whose terminals were encapsulated to the PEEs of the sensors. The capacitance was measured using an LCR meter at a frequency of 1 kHz. The normalized change in capacitance was calculated by $\Delta C/C_0 = (C-C_0)/C_0$, where $C_0$ is the initial capacitance, and $C$ is the current capacitance.

**Tensile sensor.** Dumbbell-shaped sensors with a gauge part of dimensions of 10 mm × 2 mm × 0.8 mm were prepared by printing a dielectric layer of Tango (thickness: 0.4 mm) between two layers of PEE (thickness: 0.2 mm). The sensor was stretched uniaxially using a testing machine (Instron 5966) at a speed of 30 mm min$^{-1}$ with a maximum tensile strain of 50%. For the stability test, the sensor was stretched at a speed of 6 mm s$^{-1}$ with a fixed tensile strain of 33% for 1800 cycles. During tension, the capacitance was recorded in real-time.

**Compressive sensor.** The compressive sensors without micro-structures were printed into a circular shape of diameter 30 mm and thickness 1.5 mm (0.5 mm for each layer), with two tails for connection. A cylindrical acrylic plate of diameter 35 mm was fixed to the loading head of the testing machine and used as an indenter. During the test, the indenter gradually moved downward at a speed of 2 mm min$^{-1}$ to compress the sensor. As the displacement of the indenter increased from 0 to 0.8 mm, the compression varied from 0 to 670 kPa accordingly and the capacitance was recorded in real-time. For the stability test, the sensors were pressed at a speed of 0.8 mm s$^{-1}$ with a maximum displacement of 0.8 mm for 10000 cycles. The compressive sensors with microstructures were printed into a circular shape with a diameter of 20 mm and a thickness of 3.0 mm. The PEE layers were 0.5 mm thick and the DE layer was printed into many columnar structures with a diameter of 0.8 mm and a height of 2 mm. During the test, the indenter gradually moved downward at a speed of 0.5 mm min$^{-1}$ with a maximum displacement of 2 mm.

**Shear sensor.** Three types of shear sensors with identical geometries (20 mm × 10 mm × 1.5 mm) but different patterns were designed and 3D printed. To ensure symmetry, the top and bottom surfaces of the sensor were glued to acrylate plates, which were then clamped to the testing machine. The sensors were sheared at a speed of 2 mm min$^{-1}$ with a maximum shear displacement of 1 mm, corresponding to a shear strain of 66.7%. For the stability test, the shear sensors with a flat front line were sheared cyclically at a speed of 0.8 mm s$^{-1}$ with a maximum shear displacement of 0.8 mm for 5000 cycles.

**Torsional sensor.** Two kinds of torsional sensors were printed and connected to a self-built LabVIEW-controlled torsional loading system (Fig. S29). The sensors were twisted at an angular velocity of 3.6° s$^{-1}$. For the stability test, the bisected sensor was twisted at an angular velocity of 36° s$^{-1}$ with a maximum angle of 30° for 5000 cycles. The capacitance was measured at 100 kHz.

**Integrated sensors.** (i) Integrated tensile and compressive sensor. For the single-mode test of tension, the sensor was cyclically stretched to 25%, 50%, and 100% strain using a universal testing machine (Instron 5966) at a speed of 0.5 mm s$^{-1}$, each for 10 cycles. For the single-mode test of compression, a rectangular acrylic plate (10 mm × 8 mm) was used as an indenter to apply cyclic pressure to the right part of the sensor at a speed of 2 mm min$^{-1}$ with compressive strains of 15%, 25%, and 33%, each for 10 cycles. For the combined deformation test, the sensor was stretched to 25%, 50%, or 100%, fixed, and then subject to a

compression varying from 0 to 771 kPa. During all mechanical loadings, the capacitances of $C_1$ and $C_2$ were measured synchronously. (ii) Integrated compressive and shear sensor. The capacitance of the sensor was measured using an LCR meter (TH2838A, Changzhou Tonghui Electronic Co. Ltd, China) at a testing frequency of 1.2 kHz. Both the compressive test and shear test were performed using a mechanical testing machine (Instron 68SC). Two acrylic plates were glued to the top and bottom surfaces of the sensor as fixtures. During the compressive test, the machine compressed the sensor along the thickness direction at a loading velocity of 0.5 mm min$^{-1}$. During the shear test, the machine pulled the acrylate plate along the in-plane direction at a loading velocity of 1 mm min$^{-1}$. (iii) Integrated torsional and compressive sensor. The capacitance of the sensor was measured using an LCR meter (TH 2838 A) at a testing frequency of 1.2 kHz. Two acrylic plates were glued to the top and bottom surfaces of the sensor as fixtures. The compressive test was performed using a mechanical testing machine (Instron 68SC) at a loading velocity of 0.5 mm min$^{-1}$. The torsional test was performed using the self-built LabVIEW-controlled torsional loading system (Fig. S29) at an angular velocity of 5° s$^{-1}$.

**Remote-control unit.** The five sensors of the remote-control unit shear the same ground electrode and are separately connected to five channels of a multiple relay, which is then connected to an LCR meter (TH2838A). The relay receives and processes one signal once at a time and its on-off state is regulated by the digital signals sent from an Arduino UNO board. The LCR meter operates at medium speed with a frequency of 1 kHz and a voltage of 0.5 V. The real-time capacitances and the normalized capacitances of the five sensors are recorded by a LabVIEW program (National Instruments, Austin, TX, USA), and are displayed on the computer screen. After the program gets started, the capacitance of each sensor is measured 20 times and then the average value is taken as the initial capacitance. Subsequently, the capacitance increases upon the loading of a finger. Once the normalized capacitance $C/C_0$ of a sensor is larger than the prescribed threshold, the sensor is switched on and a corresponding signal is generated, which eventually leads to an operation command for the drone.

**Adhering a printed sensor and a fabric using a hydrogel adhesive**
A hydrogel solution was prepared using acrylamide (AAM) as the monomer, 0.625 mol% PEGDA as the crosslinker, and 5 wt% TPO as the photo-initiator. AAM, PEGDA, and TPO were dissolved in deionized water with a water content of 80 wt% to form a transparent precursor as the printing ink. We first printed the sensor with a dimension of 50 mm × 10 mm × 1.5 mm (0.5 mm for each layer) following the same steps as before. Then we printed a layer of the hydrogel with a thickness of 1 mm as an adhesive layer on the sensor. After printing, the printed structure was subjected to UV light irradiation with 365 nm wavelength for 1 h for complete curing of the printed structure. For the 180° peeling test, a fabric was bonded onto the hydrogel and the test was performed on an Instron 5966 with a 100 N load cell at a speed of 10 mm min$^{-1}$.

**Response time**
To test the response times of the shear sensor and the compressive sensor of the remote-control unit, the LCR meter was set to the fast mode with a testing frequency of 3 kHz. An instantaneous load was applied and removed, and the real-time capacitance was measured by the LCR meter. Due to the limitation of the measuring circuits, the response times were measured to be about 52 ms at various loadings for both the shear sensor and the compressive sensor (Fig. S30).

**Finite element analysis**
FEA was performed using the commercial package COMSOL Multiphysics 6.0. According to the experimental data, both the PEE and DE

were modeled as incompressible neo-Hookean materials with shear moduli of 90.7 kPa and 294 kPa, respectively. DE was modeled as a linear dielectric material with relative dielectric constant $\varepsilon_r = 3$, and PEE was simplified as an equipotential body. The deformation caused by electrostatic force was ignored.

**Tensile sensor.** The upper PEE was applied with a voltage of 1 V and the lower PEE was grounded. For boundary conditions, the left side of the sensor was fixed and the right side was subjected to a specified displacement $L$. The maximum value of the specified displacement $L$ was set to be 10 mm.

**Integrated tensile and compressive sensor.** The bottom-right PEE was grounded and both the top-right and bottom-left PEEs were applied with 1 V. For compression, the bottom-right PEE was fixed and the top-right PEE was compressed by a displacement $L_1$. For tension, the bottom-right PEE was fixed and the bottom-left PEE was elongated by a displacement $L_2$. $L_1$ and $L_2$ are 0.75 mm and 1 mm for the rationally designed sensor, and $L_2$ is 2.5 mm for the badly designed sensor.

**Remote-control system for a drone**
A remote-control unit consisting of one compressive sensor and four shear sensors was designed and 3D printed. A copper wire was inserted into the cylindrical PEE($C_5$) during the printing process. Four copper wires were fixed to each side of the remote-control unit with PEE precursor. The five sensors of the remote-control unit shared the same ground electrode and were separately connected to five channels of a multiple relay, which was then connected to an LCR meter (TH2838A, Changzhou Tonghui Electronic Co. Ltd, China). The relay received and processed one signal once at a time and its on-off state was regulated by the digital signals sent from an Arduino UNO board. The output end of the Arduino UNO board was connected to five channels of another multiple relay, which was further connected to the corresponding pins of the PCB of the drone controller by welding. The LCR meter operated at medium speed with a frequency of 1 kHz and a voltage of 0.5 V. The real-time capacitances and the normalized capacitances of the five sensors were measured in a loop one after another by an LCR meter and were recorded by a LabVIEW program (National Instruments, Austin, TX, USA), and were displayed on the computer screen. After the program got started, the capacitance of each sensor was measured 20 times and then the average value was taken as the initial capacitance. Subsequently, the capacitance increased upon the loading of a finger. Once the normalized capacitance $C/C_0$ of a sensor was larger than the prescribed threshold, the corresponding pin was switched on and a corresponding signal was generated, which eventually lead to an operation command for the drone. During operation, the remote-control unit was worn on the hand back and connected to a controlling circuit containing two relays, an LCR meter, a LabVIEW controlling program, and an Arduino board (Fig. S31). In response to the perturbation of a finger, the capacitance of a sensor increased. When the value of $C/C_0$ exceeded a threshold, the circuit was switched on and a corresponding steering order was sent to the drone. The thresholds for capacitors $C_1$, $C_2$, $C_3$, and $C_4$ were set to be 1000 and the threshold for $C_5$ was set to be 1.1. The order of "flip" was a pre-order that the signal of $C_5$ should be followed by another order of one of the other four sensors, such that the drone would flip in the corresponding direction. The drone executed the orders accurately. The source file of the LabVIEW program used to control the remote-control system was uploaded as a supplementary material.

**Reporting summary**
Further information on research design is available in the Nature Portfolio Reporting Summary linked to this article.

## Data availability

The data that support the findings of this study are available from the corresponding authors upon request. The LabVIEW program for wirelessly controlling the flight of a drone is available at https://doi.org/10.6084/m9.figshare.23807529.

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

## Acknowledgements

The work at the Southern University of Science and Technology (SUS-Tech) is supported by the Science, Technology, and Innovation Com-mission of Shenzhen Municipality (ZDSYS20210623092005017), the Natural Science Foundation of Guangdong Province (2022A1515010601), the Stable Support Plan Program of Shenzhen Natural Science Fund Grant (20200925174603001), and the Centers for Mechanical Engineer-ing Research and Education at MIT and SUSTech. The authors acknowl-edge the assistance of SUSTech Core Research Facilities.

## Author contributions

Q.G. and C.Y. conceived the idea and designed the study. C.L., J.C., Y.H. and X.H. prepared the samples and conducted the experiments. C.L., J.C. and Y.H. analyzed the data with the help from Z.X., C.L., J.C. and Y.H. prepared the figures. C.Y. drafted the manuscript with the inputs from

other authors. All authors commented on the manuscript. Q.G. and C.Y. supervised the study.

## Competing interests

The authors declare no competing interests.

## Informed consent

Informed consent was obtained from all human research participants for the skin adhesive tests.
