## [Peer Review File · Nature Communications]

Reviewers' comments:

Reviewer #1 (Remarks to the Author):

Ge et al reported an ionotronic sensor strategy by using DLP-based multi-material 3D printing with a printable precursor of polyelectrolyte elastomer. The leakage-free polyelectrolyte elastomer contributed to the extraordinary stability and the multi-material 3D printing capability allows high flexibility in the structural design. A wearable remote-control unit for a drone was demonstrated. The topic is interesting, however, this manuscript didn't meet the merits to be published in Nat Comm at current state, some critical defects are shown as following:

1. The authors do not clearly discuss what we have learned from this work about general material design and how we can use their results to make structures with desired prescribed properties. Comments about these issues appear here and there in the paper, but I believe there should be a designated paragraph or two at the conclusion section addressing this extensively. Some related questions are, why did the authors choose the specific materials? how the printing parameters are chosen and optimised? How about the interface? How about the robustness of the printed structure in terms of the solvent resistance and thermal durability? ...The chemical – physical assessments on the printed structure seem incomplete. Lost queries remains on the surface property, durability, etc.

2. The figure contents need be polished, there are lots of issues as,

2.1 In Fig1e, the tensile curve seems not be well presented, I am not sure the claim in line 141 of 156 ± 19 kPa, 799 ± 57 kPa, and 526 ± 33 kPa is applaudable;

2.2 In Fig1f, the testing method needs a diagram to illustrate the process with defined parameters;

2.3 In Fig.2, the structure – capacitive property relationship has not been clearly defined with the structural variables. In Fig 2d, the pressure seems large, not sure this will bring any advantage in the applications.

3. The demonstration is lack of lots of quantitated details, the responding time has not been well examined. The mechanical input based signal generation seems very slim, not mention that the other electronic units are commercial based. A lot of capacitive based Human-Machine Interface can be demonstrated, the current demonstration seems not the best to show the advantage for this work.

4. Another major flaw is that the manuscript is full of typos, mistakes in both grammar and syntax, and in many cases incomprehensible. I will not enumerate all the issues here, but I advise the authors to carefully edit their paper before resubmitting to avoid sloppiness.

In conclusion, the quality of manuscript doesn't meet the merits for publishing in Nat Comm. It might fit more to some specified journal i.e. Sci. rep. etc. I would recommend to reject or transfer.

Reviewer #2 (Remarks to the Author):

In this manuscript, Li and coauthors reported ionotronic sensors fabricated by multi-materials 3D printing technologies. They show different type of sensors such as tensile, pressure, shear, and torional sensors. They also show the demonstrate a multi-functional ionotronic sensor that can decipher the signals of compression, tension, or their combination. Finally they demonstrate a wearable remote-control unit that can wirelessly communicate with a drone. The claim of this manuscript is that the multi-material 3D printing allows high flexibility in the structural design of the sensors, enabling the sensing of multiple modes of mechanical stimuli, and that the polyelectrolyte elastomer is stretchable, conductive, and resistant to ion leakage, contributing to the extraordinary durability of the sensors. The results are well organized and enough good to be published; however, the reviewer requests major revision.

- 1) Authors seem to solely compare the functionality and properties with other ionotronic sensors. However, the reviewer suggests to compare the performance with other sensors using different materials since there are so many these types of sensors reported recently (some example review articles are: doi.org/10.1016/j.cej.2021.129949, doi.org/10.1002/admt.202001023), and some of them are fabricated with 3D-printing technologies (doi.org/10.1002/adma.202004782).
- 2) Related to Comment 1, the introduction should discuss the advantages of ionotronic sensors in comparison with others using different type of materials.
- 3) As shown in Figure 3, separation of multiple strain information is important. I'm curious to know whether shear and torsional sensors can separate signals of such target strains from pressure and tensile strains?
- 4) Page 10 Line 224, "which has not been realized before."; Since there are shear/ torsional sensors, the reviewer guess that the authors try to mean that "which has not been realized before using ionotronic sensors". This sentence should be revised to clarify the meaning.
- 5) Cyclic conditions are important, so the reviewer suggest to show these information clearly in figures and corresponding texts (Figure 2 h and l).
- 6) The text has some grammatical and editorial problems which hinder readers to understand the meaning of each sentence. I suggest the authros ask proof reading of the entire manuscript by a native speaker. Some examples are:
 - A) Page 3, Line 46, "Whereas...remains elusive ."; I don't understand what "remain elusive" mean. This sentence is little bit too long to understand appropriately, so I suggest to dividing into some sentences.
 - B) Page 3, Line 53, "the stability of stretchable ionic conductors aside ,": why do authors aside the stability? Normally the stability is an important issue and thereby should not be ignored. If the authors mean that the stability issue is not the focus of this manuscript, I don't think this phrase is needed here.

C) Page 4, Line 76, “despite the elaborate structures”: I don’t understand the meaning of this. The authors describe “simply print one single material” in the previous sentence, which does not correspond to the following sentence.

D) Page 4, Line 79, “whereby” is not appropriate here.

Reviewer #3 (Remarks to the Author):

This work integrated some reported works to create multi-material 3D-printed ionotronic sensors with diverse functionalities. The method to fabricate a variety of ionotronic sensors by digital light processing (DLP)-based 3D printing forms is the nearly same to the latest work presented by the authors. Furthermore, many literatures claimed that their highly stretchable sensors could be used for versatile functionalities for applications across, but what exact strain and sensitivity is needed? An additional sample demonstration should be added to clarify the diverse functionalities. So, this reviewer think this work needs to be improved to display its significance before acceptance.

Further comments are noted below:

(1) In the introduction part, the motivation of this work is not clearly explained. The manuscript should point out the exact strain, sensitivity needed in stretchable ionotronics for sensing.

(2) In this study, the authors synthesize a pair of cation and anion as polyelectrolyte elastomer without any reference. Please clarify it with more details. Please draw the whole structure of the sensor and point out the size and chemical of each layer.

(3) How to demonstrate that the PEE contains fixed anions and mobile cations? The samples should be characterized with more experimental details, such as in-situ FTIR spectra.

(4) There are many expression mistakes about the sensor properties.

(5) For the claimed application as controller system attached on human skin, more bionic experiment and cell cytotoxicity test should be provided.

REVIEWER COMMENTS

Response: the authors would like to thank the editors and reviewers for their valuable comments which help us improve our paper significantly. We have carefully addressed the reviewers' comments and suggestions point-by-point, and revised our paper accordingly. Our responses to each of the comments are as follows.

Reviewer #1:

General Comment: Ge et al reported an ionotronic sensor strategy by using DLP-based multi-material 3D printing with a printable precursor of polyelectrolyte elastomer. The leakage-free polyelectrolyte elastomer contributed to the extraordinary stability and the multi-material 3D printing capability allows high flexibility in the structural design. A wearable remote-control unit for a drone was demonstrated. The topic is interesting, however, this manuscript didn't meet the merits to be published in Nat Comm at current state, some critical defects are shown as following:

Response: We thank the reviewer for the constructive comments and suggestions. By following them, we have thoroughly revised our manuscript and significantly improve the quality of our work. Therefore, we sincerely ask the reviewer to re-evaluate the possibility of our work to be published in Nature Communications.

Comment 1.1: The authors do not clearly discuss what we have learned from this work about general material design and how we can use their results to make structures with desired prescribed properties. Comments about these issues appear here and there in the paper, but I believe there should be a designated paragraph or two at the conclusion section addressing this extensively. Some related questions are, why did the authors choose the specific materials? how the printing parameters are chosen and optimised? How about the interface? How about the robustness of the printed structure in terms of the solvent resistance and thermal durability? ...The chemical – physical assessments on the printed structure seem incomplete. Lost queries remains on the surface property, durability, etc.

Response: We thank the reviewer for the constructive suggestions. The motivation of our work is that existing ionotronic sensors usually can only sense a single mode of deformation such as compression or tension, which significantly restricts the scope of applications. The limitation in sensing modes is due to the limitation in device structures, which in turn is due to the deficiency in manufacturing techniques. Moreover, the employed ionic conductors suffer from leakages and greatly hamper the stability of the ionotronic sensors. In our work, we resolve the deficiencies in sensing mode and stability simultaneously by synthesizing a new type of **leakage-free polyelectrolyte elastomer** and using DLP-based **multi-material 3D printing** technique to fabricate a variety of long-term stable ionotronic sensors with multi-mode sensing capabilities. As shown in Figure R1 (Figure 1 in the revised manuscript), we have fabricated various architected ionotronic sensors with multi-mode sensing capabilities, which mimic the multi-mode sensing performances of the human skin. **We have demonstrated the fully multi-material 3D printed sensors capable of sensing tension, compression, shear, and torsion, as well as combined tension and compression, combined compression and shear, and combined torsion and compression without signal cross-talks.** This is **the first work** that reports an approach to design and manufacture **fully 3D printed ionotronic sensors** that are capable of **sensing multiple modes of deformation**.

Figure R1 (Figure 1 in the revised manuscript). Skin-mimicked ionotronic sensors with multi-mode sensing capabilities by DLP-based multi-material 3D printing. (a) Schematic of the human skin containing various mechanoreceptors. The SAI responds to touch and static pressure, the SAIL responds to stretching, the RAI responds to touch and dynamic pressure, and the RAIL responds to deep pressure and vibration. (b) Human skin is capable of multi-mode sensing, such as compression, tension, shear, and torsion, as well as combined tension and compression, combined compression and shear, and combined torsion and compression without signal cross-talks.

combined compression and shear, and combined torsion and compression, without signal interferences. (c) 3D printing of various architected ionotronic sensors for multi-mode sensing using polyelectrolyte elastomers with robust interfaces and without leakage.

For the general material design, we have rewritten the sentences about polyelectrolyte elastomers in the Introduction:

“Polyelectrolyte elastomers with either cations or anions fixed to the backbones provide a promising remedy for enduring ionotronics by simultaneously resolving the predicaments of solvent-leakage and ion-leakage.”

Furthermore, in the characterizations of PEE, we have added the synthetic strategy, the schematic of the synthetic process, and the chemical structures of ingredients used for PEE in the revised Figure 2, as shown in Figure R2. The key idea is to use an ionic monomer containing an acrylate functional group for free radical photo-polymerization such that, one type of ions will be engrafted to the polymer network and resistant to leakage and the other type of ions will be confined within the polymer network as well due to electrostatic interactions.

Figure R2 (Figure 2c-e in the revised manuscript). Schematic of the synthetic process and the network structure of PEE.

In addition, we have explained the mechanism of being resistant to ion leakage of PEE in more detail in the descriptions of the revised Figure 3h:

“When in contact with DE, the ions in the PEE tend to diffuse toward the DE due to the concentration gradient. However, the directional diffusion of the fixed anions exerts tensile stress on the polymer chains of PEE, which counteracts the chemical potential of the anions to prevent long-range diffusion (Figure 3h). Meanwhile, the directional

diffusion of the mobile cations is also prohibited due to the electrostatic interactions. Consequently, PEE is resistant to ion leakage and the PEE-based ionotronic sensor is stable.”.

We attach the schematic in Figure R3. Now we believe the general material design should be clear.

Figure R3 (Figure 3h in the revised manuscript). Schematics illustrating the prohibition of ion leakage in the PEE-based sensor.

As for how to make the structures with desired prescribed properties, the multi-material 3D printing capability allows one to rationally design the structures of the sensor with high flexibility. For example, since the capacitance change of the compressive sensor is inversely proportional to the thickness of DE layer, and the sensitivity is inversely proportional to the stiffness of DE layer, we can increase the sensitivity of the sensor by introducing microstructures to the DE layer, as shown in Figure R4.

Figure R4 (Figure 4d in the revised manuscript). Variations of $\Delta C/C_0$ with pressure for the compressive sensors with/without microstructures. The inset shows the snapshot image of a micro-structured sensor with a diameter of 20 mm.

As another example, since the capacitance change of the shear sensor for a differential increment in shear displacement dl is proportional to the effective overlap area of the two PEE layers, given by

$$\Delta A = \int w(l)dl,$$

where w is the characteristic dimension in the width direction, we can tailor the sensitivity by programming the pattern of the front line. As shown in Figure R5, we have designed three patterns for the front line and obtained three sensitivities. When the front line is flat, $w(l)$ is a constant and the change in capacitance is proportional to the shear displacement to the first power,

$$\Delta C \propto \Delta A \propto \Delta l.$$

When the front line is zig-zag, $w(l)$ is proportional to the shear displacement to the first power, and the change in capacitance is proportional to the shear displacement to the second power,

$$\Delta C \propto \Delta A \propto (\Delta l)^2.$$

When the front line is parabolic, $w(l)$ is proportional to the shear displacement to the second power, and the change in capacitance is proportional to the shear displacement to the third power,

$$\Delta C \propto \Delta A \propto (\Delta l)^3.$$

Figure R5. Shear sensors with tailorable sensitivities. (a) Structure and principle of the shear sensor (with a flat front line). Schematics showing the design of the shear sensors with (b) a zigzag pattern and (c) a parabolic pattern. (d) Variation of $\Delta C/C_0$ with shear strain for the shear sensors with different profiles as indicated.

As the third example, since the capacitance change of the torsional sensor for a differential increment in twist angle is proportional to the effective overlap area of the two PEE layers, given by

$$\Delta A = n\Delta\theta r^2/4,$$

where n is the number of division and r is the radius of the sensor, we can tailor the sensitivity by programming the pattern of the PEEs. As shown in Figure R6, we have designed two patterns and obtained two sensitivities.

Figure R6. Torsional sensors with tailorable sensitivities. (a) Structure and principle of the torsional sensor (bisected). Schematics showing the design of the shear sensors with (b) bisection and (c) quartering pattern. (d) Variation of $\Delta C/C_0$ with twist angle for the torsional sensors with different designs as indicated.

Moreover, as shown in Figure R7, we have designed and fabricated three types of integrated ionotronic sensors that can sense combined deformations without mutual signal interference. For each design, we have explained the designing principles and the details can be found in the corresponding texts in the revised manuscript. **To sum up, both the general material design and the structure-property relationships have been explained in detail in the revised manuscript.**

Figure R7 (Figure 5 in the revised manuscript) Integrated ionotronic sensors that can sense different stimuli without mutual signal interferences. (a) Design and principle of the integrated tensile and compressive sensor. Capacitor C_1 measures compression and capacitor C_2 measures tension. (b) Snapshot image of a printed sensor. (c) Equivalent circuit diagram of the sensor. (d) $\Delta C/C_0$ of C_1 and C_2 when the compressive unit or the tensile unit is activated. Responses of the sensor subjected to 10 cycles of (e) compression, and (f) tension with various strains as indicated. (g) The signal maps of C_1 and C_2 under the combined deformation of compression and tension. (h) Design and principle of the integrated compressive and shear sensor. Capacitor C_3 measures compression and capacitor C_4 measures shear. (i) Snapshot images of a printed sensor. (j) Equivalent circuit diagram of the sensor. (k) $\Delta C/C_0$ of C_3 and C_4 when the compressive unit or the shear unit is activated. (l) Design and principle of the integrated torsional and compressive sensor. Capacitor C_5 measures compression and capacitor C_6 measures torsion. (m) Snapshot image of a printed sensor. (n) Equivalent circuit diagram of the sensor. (o) $\Delta C/C_0$ of C_5 and C_6 when the compressive unit or the torsional unit is activated.

For the specific materials, we have stated in the Introduction about the general principle of polyelectrolyte elastomers:

“Polyelectrolyte elastomers with either cations or anions fixed to the backbones provide a promising remedy for enduring ionotronics by simultaneously resolving the predicaments of solvent-leakage and ion-leakage.”

The key idea is to use an ionic monomer containing an acrylate functional group for free radical photo-polymerization such that, one type of ions will be engrafted to the polymer network and resistant to leakage, and another type of ions will be confined within the polymer network as well due to electrostatic interactions. Although the principle is generic, one has to choose specific chemistries for deployments. In our work, following the general principle of polyelectrolyte elastomers, **we synthesize a photo-curable ionic monomer BS (Figure R8a-b) and then synthesize the PEE through photo-polymerization (Figure R8c)**. We have added detailed information about the material system used in our work in the revised manuscript, including the synthetic strategy, the schematic of the synthetic process, and the chemical structures of the ingredients used for PEE, as shown in Figure R8.

Figure R8 (Figure S1 in Supplementary Materials and Figure 2a-e in the revised manuscript). Synthesis of monomer and PEE. (a) Schematic of the synthetic process of the photo-curable ionic monomer and the PEE. (b) Synthesis of the monomer, BS. The image shows the transparent liquid-state BS. (c) Synthesis of the PEE with chemical structures shown.

For the printing parameters and optimization, we have conducted more studies regarding the in-situ photo-rheological behaviors and the energy density-layer thickness relationships (as shown in Figure R9), and added a representative printed multi-material grid structure to the revised manuscript. We have added the following sentences to Page 9 in the revised manuscript:

“We perform in-situ photo-rheological characterizations to investigate the photo-reactivity of PEE and DE. As shown in Figure 3a, we identify the gelation time when the storage modulus curve intersects the loss modulus curve. To cure a 50 μm thick layer, the gelation time of DE is ~ 2 s and the gelation time of PEE is ~ 11 s, indicating that both materials are highly photocurable. Moreover, we carry out photo-rheological

characterizations (Figure S9) to determine the required gelation time (or energy density) to cure PEE or DE samples with different layer thicknesses. As shown in Figure 3b, to cure a thicker layer needs a longer gelation time. Specifically, owing to the high photo-sensitivity, the curing times for 300 μm thick samples are 45.8 s for PEE (under the irradiation of 405 nm UV projection at 384.7 mJ cm^{-2}) and 20.6 s for DE (under the irradiation of 405 nm UV projection at 173.0 mJ cm^{-2}). In addition, we further test the dual-material printability between PEE and DE by printing a grid pattern board where the width of transparent DE line is 100 μm and the light-yellow PEE blocks are embedded in the DE grids (Figure 3c).”

Figure R9 (Figure 3a-c in the revised manuscript and Figure S9 in Supplementary Materials). Synthesis and characterizations of the printing of PEE.

For the interface, we have characterized the interfacial toughness by measuring the adhesion energy, as shown in Figure R10. For more clarity, we add more details about the cohesive rupture of the printed sample and the adhesive rupture of the assembled sample to the revised Figure 3:

“We perform the 180° peeling tests to assess the interfacial toughness of 3D printed and manually assembled PEE-DE bilayers (Figure 3d). The adhesion energies, given

by the plateaued normalized force, $2F_{ss}/W$, where F_{ss} is the steady-state peel force and W is the width of the sample, are 339.3 J m^{-2} and 4.1 J m^{-2} for printed and assembled samples, respectively. Cohesive rupture occurs during the peeling of the printed bilayer that PEE residues are left on the surface of DE after peeling (Figure 3e), meaning that the interface is tougher than the bulk of PEE. The strong interfacial bonding is mainly ascribed to the topological entanglements due to the similar chemistries between PEE and DE and the covalent interlinks due to the partial curing of each printing layer. By contrast, the manually assembled bilayer experiences adhesive rupture with low adhesion energy (Figure 3f).”

Figure R10 (Figure 3d-f in the revised manuscript). Peeling tests of printed and assembled PEE/DE bilayers. (a) 180° peeling curves for 3D-printed (red) and manually assembled (blue) PEE/DE bilayers. Schematics showing (b) the cohesive rupture of printed PEE/DE bilayer, and (c) the adhesive rupture of assembled PEE/DE bilayer.

For thermal durability, we have performed TGA measurements. TGA reveals that the PEE is thermally stable up to 275°C , as shown in Figure R11, which should be sufficient for most engineering applications.

Figure. R11 (Figure 2j in the revised manuscript). TGA measurement of p(BS-co-MEA) with the molar ratio of BS:MEA=1:1.

For testing solvent resistance, we soaked samples in water for 7 days. After taking out and exsiccating the samples, we measured the changes in mass and conductivity. For comparison, we also performed control experiments using a LiTFSI doped elastomer (abbreviated as LiE). The PEE maintains weight and conductivity well whereas the LiE dramatically loses weight and conductivity, by 26.6% and 98.4%, respectively, as shown in Figure R12.

Figure R12 (Figure 2k in the revised manuscript). The changes in weight (orange) and conductivity (green) of PEE and LiE after soaking in water for 7 days and drying.

We also tested the solvent stability in organic solvent. As shown in Figure R13, soaking in methyl cyanide (MeCN) leads to similar results that the weight and conductivity change negligibly for PEE but enormously for LiE. Besides, the PEE maintains shape well while the LiE becomes ragged after the test. These results have been added to the supplementary materials.

Figure R13 (Figure S7 in Supplementary Materials). Stability of PEE and LiE in water. (a) The changes in weight and conductivity after soaking in water for 7 days and drying. (b) Images of LiE and PEE samples before and after the test.

Furthermore, we have carried out more assessments for the material and the printed sensors, including the FTIR spectrum and transmittance measurements, cytotoxicity tests, and response time of the sensing elements of the control unit, as shown in Figure R14. The details can be found in the revised manuscript and revised supplementary materials.

Figure R14 (Figure 2f, 2i in the revised manuscript and Figure S8, S22 in Supplementary Materials) (a) FTIR spectra of the precursor of p(BS-co-MEA) before and after polymerization. The inset shows the disappearance of the peak corresponding to the vinyl group. (b) Transmittance of various p(BS-co-MEA). (c) Live assay after 24 hours post seeding of NIH-3T3 cells cultured with PEE and DE slabs. Response time tests of (d) shear sensor and (e) compressive sensor of the remote-control unit.

Comment 1.2: The figure contents need be polished, there are lots of issues as,

Comment 1.2.1: In Fig1e, the tensile curve seems not be well presented, I am not sure the claim in line 141 of 156 ± 19 kPa, 799 ± 57 kPa, and 526 ± 33 kPa is applaudable;

Response: In original Figure 1e, each solid blue curve is averaged from at least three parallel samples, and the shadows represent the standard deviations. We have moved this figure to the revised supplementary materials and added the stress-strain curves of PEE with

different compositions in the revised main Figure 2g, as shown in Figure R15. The expressions such as 156 ± 19 kPa are commonly used to represent the average values of mechanical properties such as Young's modulus with the information of standard deviations.

Figure R15 (Figure 2g in the revised manuscript) Nominal uniaxial tensile stress-strain curves of p(BS-co-MEA) with various molar ratios of BS:MEA.

Comment 1.2.2: In Fig1f, the testing method needs a diagram to illustrate the process with defined parameters;

Response: We thank the reviewer for the constructive suggestion. We have split the original Figure 1f into three sub-figures in the revised Figure 3, as shown in Figure R16. We have described the peeling test and discussed the results on Page 10 in the revised manuscript:

“We perform the 180° peeling tests to assess the interfacial toughness of 3D printed and manually assembled PEE-DE bilayers (Figure 3d). The adhesion energies, given by the plateaued normalized force, $2F_{ss}/W$, where F_{ss} is the steady-state peel force and W is the width of the sample, are 339.3 J m^{-2} and 4.1 J m^{-2} for printed and assembled samples, respectively. Cohesive rupture occurs during the peeling of the printed bilayer that PEE residues are left on the surface of DE after peeling (Figure 3e), meaning that the interface is tougher than the bulk of PEE. The strong interfacial bonding is mainly ascribed to the topological entanglements due to the similar chemistries between PEE and DE and the covalent interlinks due to the partial curing of each printing layer. By

contrast, the manually assembled bilayer experiences adhesive rupture with low adhesion energy (Figure 3f).”.

Figure R16 (Figure 3d-f in the revised manuscript). Peeling tests of printed and assembled PEE/DE bilayers. (a) 180° peeling curves for 3D-printed (red) and manually assembled (blue) PEE/DE bilayers. Schematics showing (b) the cohesive rupture of printed PEE/DE bilayer, and (c) the adhesive rupture of assembled PEE/DE bilayer.

For more clarity, we have added a paragraph to describe the details of peeling test in the Experimental Section in the revised manuscript:

“180° peel test was performed to measure the adhesion energy between PEE and DE, using the Instron 5966 with a 100N load cell at a constant peeling speed of 30 mm min⁻¹. A stiff backing was bonded to each layer of samples using double-sided mesh tape. For the printed sample, the sample was printed with dimensions of 60 mm × 10 mm × 1 mm (0.5 mm for each layer) and a 10 mm long pre-crack was made. For the assembled sample, the sample was prepared by assembling a piece of PEE (60 mm × 10 mm × 0.5 mm) and a piece of DE (60 mm × 10 mm × 0.5 mm) and then the sample was stood for 30 minutes before the test.”

Comment 1.2.3. In Fig.2, the structure – capacitive property relationship has not been clearly defined with the structural variables. In Fig 2d, the pressure seems large, not sure this will bring any advantage in the applications.

Response: We have revised the structure – capacitive property relationship both in the revised Figure 4 and in our response to **Comment 1.1**. In particular, whereas the key innovation of our work is the multi-mode sensing capabilities of the printed ionotronic sensors, we have introduced microstructures to the DE layer of the compressive sensor,

improved its sensitivity by two orders of magnitude, and reduced the magnitude of pressure to the order of 10 kPa, as shown in Figure R17.

Figure R17 (Figure 4d in the revised manuscript). Variations of $\Delta C/C_0$ with pressure for the compressive sensors with/without microstructures. The inset shows the snapshot image of a micro-structured sensor with a diameter of 20 mm.

Comment 1.3: The demonstration is lack of lots of quantitated details, the responding time has not been well examined. The mechanical input based signal generation seems very slim, not mention that the other electronic units are commercial based. A lot of capacitive based Human-Machine Interface can be demonstrated, the current demonstration seems not the best to show the advantage for this work.

Response: We thank the reviewer for the constructive suggestions. We have added the original Figure S13 to the manuscript in the revised Figure 6. Furthermore, we have characterized the response times of the shear sensor and the compressive sensor of the control unit, and added them to the revised supplementary materials in Figure S22, as shown in Figure R18. The response times are measured to be 52 ms for the shear sensor and the compressive sensor, respectively, within the capabilities of our measuring instruments.

Figure R18 (Figure S22 in Supplementary Materials). Response time tests of (a) shear sensor and (b) compressive sensor of the remote-control unit.

Comment 1.4: Another major flaw is that the manuscript is full of typos, mistakes in both grammar and syntax, and in many cases incomprehensible. I will not enumerate all the issues here, but I advise the authors to carefully edit their paper before resubmitting to avoid sloppiness.

Response: We thank the reviewer for the kind suggestions. We have carefully and thoroughly polished the text.

Comment 1.5: In conclusion, the quality of manuscript doesn't meet the merits for publishing in Nat Comm. It might fit more to some specified journal i.e. Sci. rep. etc. I would recommend to reject or transfer.

Response: We thank the reviewer again for the constructive comments and suggestions. By following them, we have thoroughly revised our manuscript and significantly improve the quality of our work. Therefore, we sincerely ask the reviewer to re-evaluate the possibility of our work to be published in Nature Communications.

Reviewer #2:

General Comment: In this manuscript, Li and coauthors reported ionotronic sensors fabricated by multi-materials 3D printing technologies. They show different type of sensors such as tensile, pressure, shear, and torional sensors. They also show the

demonstrate a multi-functional ionotronic sensor that can decipher the signals of compression, tension, or their combination. Finally they demonstrate a wearable remote-control unit that can wirelessly communicate with a drone. The claim of this manuscript is that the multi-material 3D printing allows high flexibility in the structural design of the sensors, enabling the sensing of multiple modes of mechanical stimuli, and that the polyelectrolyte elastomer is stretchable, conductive, and resistant to ion leakage, contributing to the extraordinary durability of the sensors. The results are well organized and enough good to be published; however, the reviewer requests major revision.

Response: We appreciate the reviewer for the comprehensive summary and positive remarks on our work. In the following, we have addressed the reviewer's comments point-by-point.

Comment 2.1: Authors seem to solely compare the functionality and properties with other ionotronic sensors. However, the reviewer suggests to compare the performance with other sensors using different materials since there are so many these types of sensors reported recently (some example review articles are: doi.org/10.1016/j.cej.2021.129949, doi.org/10.1002/admt.202001023), and some of them are fabricated with 3D-printing technologies (doi.org/10.1002/adma.202004782).

Response: We thank the reviewer for the kind suggestions. In the revision, **we have cited the three suggested papers.** Indeed, there are many flexible/stretchable sensors based on electronic conductors and ionotronic sensors actually benefiting from them. For example, the abundance of electronic sensors offers enormous inspiration for the design of ionotronic sensors. Whereas we would like to keep our work more focused, we rewrite the sentences discussing the selection of capacitive sensor in the second paragraph of the Introduction and in the first paragraph of the Results and Discussions section as follows:

“Ionotronic sensors, featuring extraordinary softness, flexibility/stretchability and optical transparency, are emerging for tactile perception. Among the various types of

ionotronic sensors, the capacitive type prevails owing to the ultrahigh sensitivity, high signal-to-noise ratio, low power consumption, and less sensitive to signal drifting.”

“In general, ionotronics sensors are softer and more stretchable than their electronic counterparts. In addition, since ionotronic sensors also employ ions as the charge carrier, they potentially provide a more seamless interface with the biological systems.”

Comment 2.2: Related to Comment 1, the introduction should discuss the advantages of ionotronic sensors in comparison with others using different type of materials.

Response: We thank the reviewer for the constructive suggestion. Ionotronic sensors employ ionic conductors as the functional materials and ionic conductors, such as ionic hydrogels, ionogels, and ionically conductive elastomers, are generally soft, stretchable, and sometimes transparent and biocompatible. Consequently, compared to their electronic counterparts, ionotronic sensors are often advantageous in terms of intrinsic softness, stretchability, optical transparency, and biocompatibility. We have stated in the first paragraph of the Introduction:

“As the most quintessential stretchable ionic conductors, gel materials, including hydrogels containing dissolved salts and ionogels, are advantageous in terms of intrinsic softness, stretchability, optical transparency, and biocompatibility.”

Whereas we would like to keep the Introduction section more focused, for more comparison, we have added the following sentences to the first paragraph of the Results and Discussions section in the revised manuscript:

“In general, ionotronics sensors are softer and more stretchable than their electronic counterparts. In addition, since ionotronic sensors also employ ions as the charge carrier, they potentially provide a more seamless interface with the biological systems.”

Comment 2.3: As shown in Figure 3, separation of multiple strain information is important. I'm curious to know whether shear and torsional sensors can separate signals of such target strains from pressure and tensile strains?

Response: We thank the reviewer for the constructive suggestions. Yes, 3D printing allows the design and fabrication of integrated ionotronic sensors that can separate shear and torsion from compression. In addition to the integrated tensile and compressive sensor, we have added two more examples to demonstrate these performances to the revised Figure 5, as shown in Figure R19. One is an integrated compressive and shear sensor that can sense compression, shear, or their combination without signal cross-talk. Another is an integrated torsional and compressive sensor that can sense compression, torsion, or their combination without signal cross-talks.

Figure. R19 (Figure 5 in the revised manuscript). Integrated ionotronic sensors that can sense different stimuli without mutual signal interferences. (a) Design and principle of the integrated tensile and compressive sensor. Capacitor C_1 measures compression and capacitor C_2 measures tension. (b) Snapshot image of a printed sensor. (c) Equivalent circuit diagram of the sensor. (d) $\Delta C/C_0$ of C_1 and C_2 when the compressive unit or the tensile unit is activated. Responses of the sensor subjected to 10 cycles of (e) compression, and (f) tension with various strains as indicated. (g) The signal maps of C_1 and C_2 under the combined deformation of compression and tension. (h) Design and principle of the integrated compressive and shear sensor. Capacitor C_3 measures compression and capacitor C_4 measures shear. (i) Snapshot images of a printed sensor. (j) Equivalent circuit diagram of the sensor. (k) $\Delta C/C_0$ of C_3 and C_4 when the compressive unit or the shear unit is activated. (l) Design and principle of the integrated torsional and compressive sensor. Capacitor C_5 measures compression and capacitor C_6 measures torsion. (m) Snapshot image of a printed sensor. (n) Equivalent circuit diagram of the sensor. (o) $\Delta C/C_0$ of C_5 and C_6 when the compressive unit or the torsional unit is activated.

Comment 2.4: Page 10 Line 224, “which has not been realized before.”; Since there are shear/ torsional sensors, the reviewer guess that the authors try to mean that “which has not been realized before using ionotronic sensors”. This sentence should be revised to clarify the meaning.

Response: We thank the reviewer for the rigorous reading. Yes, we mean that “which has not been realized before using ionotronic sensors”. We have rewritten the first sentence in the last paragraph on Page 13 as follows:

“3D printing enables the fabrication of ionotronic sensors with unusual configurations to detect shear and torsion.”.

Comment 2.5: Cyclic conditions are important, so the reviewer suggest to show these information clearly in figures and corresponding texts (Figure 2 h and l).

Response: We thank the reviewer for the constructive suggestions. We have rearranged the figure (now Figure 4) to clearly show the performances of the shear sensor and the torsional sensor under cyclic loading conditions, as shown in Figure R20.

Figure. R20 (Figure 4e, 4h, 4i, 4l in the revised manuscript). Cyclic tests of shear and torsional sensors. Performances of (a) the shear sensor and (b) the torsional sensor under cyclic loading conditions.

As for the corresponding texts, we have rewritten the sentences to include more details.

For the shear sensor:

“Subject to a cyclic shear test with a maximum shear strain of 66.7%, both electrical responses (Figure 4h) and mechanical responses (Figure S16) of the shear sensor remain stable up to 5000 cycles.”

For the torsional sensor:

“Subject to a cyclic torsional test with a maximum twist angle of 30°, the sensor with a bisection pattern maintains excellent stability for 5000 cycles (Figure 4l).”

Comment 2.6: The text has some grammatical and editorial problems which hinder readers to understand the meaning of each sentence. I suggest the authors ask proof reading of the entire manuscript by a native speaker. Some examples are:

A) Page 3, Line 46, “Whereas...remains elusive .”; I don’t understand what “remain elusive” mean. This sentence is little bit too long to understand appropriately, so I suggest to dividing into some sentences.

Response: We have rewritten the sentence to be more concise as follows:

“The ionically conductive elastomers synthesized by dissolving lithium salt (e.g. lithium bis(trifluoromethanesulfonyl)imide (LiTFSI)) or zwitterions into elastomer matrices are immune to solvent leakage, but they are still susceptible to ion leakage. The

persistent concentration gradient between the interior and the exterior keeps driving the mobile ions to diffuse outwards when in contact with other elastomers.”

B) Page 3, Line 53, “the stability of stretchable ionic conductors aside ,”: why do authors aside the stability? Normally the stability is an important issue and thereby should not be ignored. If the authors mean that the stability issue is not the focus of this manuscript, I don’t think this phrase is needed here.

Response: We have rewritten the sentence as follows:

“Nevertheless, in addition to the stability of the materials, manufacturing of the devices has been another long-standing hurdle for the development of the field.”

C) Page 4, Line 76, “despite the elaborate structures”: I don’t understand the meaning of this. The authors describe “simply print one single material” in the previous sentence, which does not correspond to the following sentence.

Response: We have rewritten the sentence as follows:

“Moreover, the perceivable stimuli of these printed ionotronic sensors are mostly limited to compression and/or tension (Table S1).”

D) Page 4, Line 79, “whereby” is not appropriate here.

Response: We have rewritten the sentence as follows:

“The deficiency of sensing capabilities greatly hampers the applications of ionotronic sensors in engineering, e.g. soft robots and human-machine interactions, where the sensors are desired to have more sophisticated capability of sensing different mechanical stimuli such as shear, torsion, or even their combinations.”

Reviewer #3:

General comment: This work integrated some reported works to create multi-material 3D-printed ionotronic sensors with diverse functionalities. The method to fabricate a variety of ionotronic sensors by digital light processing (DLP)-based 3D printing forms

is the nearly same to the latest work presented by the authors. Furthermore, many literatures claimed that their highly stretchable sensors could be used for versatile functionalities for applications across, but what exact strain and sensitivity is needed? An additional sample demonstration should be added to clarify the diverse functionalities. So, this reviewer thinks this work needs to be improved to display its significance before acceptance.

Response: We thank the reviewer for the kind suggestions to help us improve our work for acceptance. Our work does not simply integrate some reported works. Existing ionotronic sensors usually can only sense a single mode of deformation such as compression or tension, which significantly restricts the scope of applications. The limitation in sensing modes is due to the limitation in device structures, which in turn is ascribed to the deficiency in manufacturing techniques. Moreover, the employed ionic conductors suffer from leakages and greatly hamper the stability of the ionotronic sensors. In our work, we resolve the deficiencies in sensing mode and stability simultaneously by synthesizing a new type of **leakage-free polyelectrolyte elastomer** and using the DLP-based **multi-material 3D printing** technique to fabricate a variety of long-term stable ionotronic sensors with multi-mode sensing capabilities. As shown in Figure R21 (Figure 1 in the revised manuscript), we have fabricated various architected ionotronic sensors with multi-mode sensing capabilities, which mimics the multi-mode sensing performances of the human skin. **We have demonstrated the sensing of tension, compression, shear, and torsion, as well as combined tension and compression, combined compression and shear, and combined compression and torsion without signal cross-talks.** This is the **first work** that reports an approach to design and manufacture **fully 3D printed** ionotronic sensors that are capable of **sensing multiple modes of deformations.**

Figure R21 (Figure 1 in the revised manuscript). Ionotronic sensors with multi-mode sensing capabilities by DLP-based multi-material 3D printing, mimic the multi-mode sensing performances of the human skin. (a) Schematic of the human skin containing various mechanoreceptors. The SAI responds to touch and static pressure, the SAII responds to stretching, the RAI responds to touch and dynamic pressure, and the RAII responds to deep pressure and vibration. (b) Human skin is capable of multi-mode sensing, such as compression, tension, combined compression and shear, and combined torsion and compression. (c) 3D printing of various architected ionotronic sensors for multi-mode sensing using polyelectrolyte elastomers with a robust interface and without leakage.

To achieve the above-mentioned goal, **on one hand, we have designed and synthesized a new type of polyelectrolyte elastomer** that is stretchable, transparent, ionically conductive, thermally stable, and leakage-resistant. In particular, to achieve the merits of the polyelectrolyte elastomer, we have synthesized the photopolymerizable ionic monomer, BS, by ourselves. We might not describe the material clearly enough in the original manuscript. In the revised manuscript, we perform more comprehensive characterizations for the material and devote Figure 2 for material characterizations, as shown in Figure R22.

Figure R22 (Figure 2 in the revised manuscript). Synthesis and characterizations of PEE. (a) The synthesis of monomer, BS. (b) An image of the as-prepared BS. (c) Chemical structures of monomers, BA and MEA, and crosslinker, HDDA. (d) Synthesis of p(BS-co-MEA) network via photopolymerization. (e) The p(BS-co-MEA) network contains engrafted negative charges and mobile positive charges. (f) FTIR spectra of the precursor of p(BS-co-MEA) before and after polymerization. The inset shows the disappearance of the peak corresponding to the vinyl groups. (g) Nominal uniaxial tensile stress-strain curves of p(BS-co-MEA) with various molar ratios of BS:MEA. (h) The variations of fracture strain and Young's modulus with the molar ratios of BS:MEA. (i) Transmittance of various p(BS-co-MEA). (j) TGA measurement of p(BS-co-MEA) with the molar ratio of BS:MEA=1:1. (k) The changes in weight (orange) and conductivity (green) of PEE and LiE after soaking in water for 7 days and drying.

On the other hand, DLP-based multi-material 3D printing is a well-established technique and anyone can use it to fabricate their own structures. Nevertheless, the printing protocols often need to be modified whenever a new material has been used. In our work, we modify the printing parameters to comprise the printing of the new

type of polyelectrolyte elastomer such that, the printed samples do not have much-deteriorated properties and the printed ionotronic sensors have a robust interface and decent resolution. We add Figure 3a-3c, regarding the in-situ photo-rheological properties, energy density-layer thickness relationships, and a representative printed grid structure, in the revised manuscript, as shown in Figure R23 a-c. Furthermore, we have shown the first direct experimental evidence that the LiTFSI doped elastomers are prone to ion leakage. By contrast, the polyelectrolyte elastomers having at least one type of ions fixed to the polymer network are resistant to ion leakage. The experimental results and the underlying mechanisms have been discussed in detail in the revised manuscript, as shown in Figure R23 d-f.

Figure. R23 (Figure 3a-c in the revised manuscript). Synthesis and characterizations of PEE and ionotronic sensors. (a) Storage modulus and loss modulus of PEE and DE vary with gelation time. (b) The variations of energy density with layer thickness for PEE and DE. (c) Microscopic image of PEE and DE grid pattern. The width of the grid is $100 \mu\text{m}$ and the length of the square is $500 \mu\text{m}$. (d) Variation of $\Delta C/C_0$ with time for PEE-based sensor and LiE-based sensor. Schematics illustrating (e) the prohibition of ion leakage in PEE-based sensor, and (f) the ion leakage in LiE-based sensor.

We agree with the reviewer that the exact strain and sensitivity should be clarified for specific applications. However, we would like to emphasize that the key point of our work is to realize long-term stable ionotronic sensors with multi-mode sensing

capability. Poor stability and the lack of functionalities are two central challenges in existing ionotronic devices. Our work resolves these two issues simultaneously, by synthesizing a new type of leakage-free polyelectrolyte elastomer and using the DLP-based multi-material 3D printing technique for fabrication, as discussed above in Figure R21.

By taking advantages of multi-material 3D printing, we have shown that the sensing of shear and torsion can also be realized through rational structure design. Furthermore, although pushing the limits of the sensing performances of ionotronic sensors is not the focus of our work, we would like to emphasize that, for the sake of multi-material 3D printing capability, the sensing performances are readily improved through the optimizations of structures. For example, we have embedded microstructures into the DE layer of the compressive sensor, **enhancing the sensitivity by two orders of magnitude and reducing the magnitude of the working pressure by more than one order of magnitude**, as shown in Figure R24. For the shear sensor and the torsional sensor, we have demonstrated that **the sensitivities can be tuned on-demand by programming the sensor structures**. Besides, microstructures have also been introduced into the DE layer of the compressive unit of the integrated compressive and shear sensor, as shown in Figure R25.

Figure R24 (Figure 4d in the revised manuscript). Variations of $\Delta C/C_0$ with pressure for the compressive sensors with/without microstructures. The inset shows the snapshot image of a micro-structured sensor with a diameter of 20 mm.

Figure R25 (Figure 5h, 5i in the revised manuscript). Integrated compressive and shear sensor. (a) Design and principle of the integrated compressive and shear sensor. Capacitor C_3 measures compression and capacitor C_4 measures shear. (b) Snapshot images of a printed sensor.

For diverse functionalities, in addition to the integrated tensile and compressive sensor, we have added two more examples to demonstrate the multi-mode sensing capabilities of printed ionotronic sensors. One is an integrated compressive and shear sensor that can sense compression, shear, or their combination without signal cross-talk. Another is an integrated compressive and torsional sensor that can sense compression, torsion, or their combination without signal cross-talks. The design, the example of a printed sample, the equivalent circuit, and the sensing performances of the two sensors have been added to Figure 5 in the revised manuscript, as shown in Figure R26.

Figure. R26 (Figure 5 in the revised manuscript). Integrated ionotronic sensors that can sense different stimuli without mutual signal interferences. (a) Design and principle of the integrated tensile and compressive sensor. Capacitor C_1 measures compression and capacitor C_2 measures tension. (b) Snapshot image of a printed sensor. (c) Equivalent circuit diagram of the sensor. (d) $\Delta C/C_0$ of C_1 and C_2 when the compressive unit or the tensile unit is activated. Responses of the sensor subjected to 10 cycles of (e) compression, and (f) tension with various strains as indicated. (g) The signal maps of C_1 and C_2 under the combined deformation of compression and tension. (h) Design and principle of the integrated compressive and shear sensor. Capacitor C_3 measures compression and capacitor C_4 measures shear. (i) Snapshot images of a printed sensor. (j) Equivalent circuit diagram of the sensor. (k) $\Delta C/C_0$ of C_3 and C_4 when the compressive unit or the shear unit is activated. (l) Design and principle of the integrated torsional and compressive sensor. Capacitor C_5 measures compression and capacitor C_6 measures torsion. (m) Snapshot image of a printed sensor. (n) Equivalent circuit diagram of the sensor. (o) $\Delta C/C_0$ of C_5 and C_6 when the compressive unit or the torsional unit is activated.

Further comments are noted below:

Comment 3.1: In the introduction part, the motivation of this work is not clearly explained. The manuscript should point out the exact strain, sensitivity needed in stretchable ionotronics for sensing.

Response: The motivation of our work is that existing ionotronic sensors suffer from poor stability and lack of functionalities, due to the poor stability of the employed ionic conductors and the simple device structures limited by manufacturing technique, which greatly impedes their practical applications. Our work resolves these two issues simultaneously by synthesizing a new type of leakage-free polyelectrolyte elastomer and using the DLP-based multi-material 3D printing technique to fabricate a variety of long-term stable ionotronic sensors with multi-mode sensing capabilities.

As for the exact strain and sensitivity, again, we agree with the reviewer that the exact strain and sensitivity should be clarified for specific applications. However, **the key point of our work is to realize long-term stable ionotronic sensors with multi-mode sensing capability.** Poor stability and the lack of functionalities are two central challenges in existing ionotronic devices. Our work resolves these two issues simultaneously, by synthesizing a new type of leakage-free polyelectrolyte elastomer and using the DLP-based multi-material 3D printing technique for fabrication. We have modified Figure 1 to highlight our key point in the revised manuscript.

Comment 3.2: In this study, the authors synthesize a pair of cation and anion as polyelectrolyte elastomer without any reference. Please clarify it with more details. Please draw the whole structure of the sensor and point out the size and chemical of each layer.

Response: We thank the reviewer for the constructive suggestions. Polyelectrolyte elastomers are a family of ionic conductors containing one type of ions grafted to the polymer network and another type of ions (counterions) mobile within the polymer network. We have cited the recently published Science paper about ionoelastomers (Kim, et. al, Science, 2020, 773-776.) in the second paragraph of the Results and

Discussions section. In the revised manuscript, in addition to the $^1\text{H-NMR}$ spectrum, we have added more details about the synthesis of the photo-curable ionic monomer and the PEE, and their chemical structures as shown in Figure R27.

Figure R27 (Figure S1 in Supplementary Materials and Figure 2a-e in the revised manuscript). Synthesis of monomer and PEE. (a) Schematic of the synthetic process of the photo-curable ionic monomer and the PEE. (b) Synthesis of the monomer, BS. The image shows the transparent liquid-state BS. (c) Synthesis of the PEE with chemical structures shown.

As for the structures and the sizes of sensors, detailed information can be found in Figure 4 and Figure 5, as well as Figures S14-S20. For example, we have provided the details about torsional sensors in Figure S18, as shown in Figure R28.

Figure R28 (Figure S17 in Supplementary Materials). Torsional sensors of different designs.

Comment 3.3: How to demonstrate that the PEE contains fixed anions and mobile cations? The samples should be characterized with more experimental details, such as in-situ FTIR spectra.

Response: We thank the reviewer for the constructive suggestions. We have performed in-situ FTIR and added the results as Figure 2f in the revised manuscript, as shown in Figure R29. The absorption peak corresponding to the vinyl groups, $\sim 1636\text{ cm}^{-1}$, vanishes, indicating the complete conversion of the monomers.

Figure R29 (Figure 2f in the revised manuscript). FTIR spectra of the precursor of p(BS-co-MEA) before and after polymerization. The inset shows the disappearance of the peak corresponding to the vinyl groups.

Comment 3.4: There are many expression mistakes about the sensor properties.

Response: We have carefully and thoroughly double-checked the text.

Comment 3.5: For the claimed application as controller system attached on human skin, more bionic experiment and cell cytotoxicity test should be provided.

Response: We thank the reviewer for the constructive suggestion. Since the printed controller consists of PEE and DE, we have performed cytotoxicity tests for the two constituent materials. Both materials exhibit low cell cytotoxicity, as shown in Figure R30. We add the results in Figure S8.

Figure R30 (Figure S8 in Supplementary Materials). Live assay after 24 hours post seeding of NIH-3T3 cells cultured with PEE and DE slabs.

REVIEWER COMMENTS

Reviewer #1 (Remarks to the Author):

The authors have tried to address my previous concerns; however, considerable gap remains from the scientific perspective. The author performed more characterizations. The key is to bring a clear scientific interpretation, rather to pile up the data.

Regarding to my first query, a mechanics study is mandatory to explain the sensing function of a single mode of deformation such as compression or tension. The current interpretation is far from enough. A FEA analysis can do the job, I suppose. I have second queries on the linearity of plots in Fig. R5 and Fig. R6. The trends appear to me that they are more link segment of a nonlinear curve, the current fit seems very rough.

For the photo-curing assessment, there is a concern on the gradient caused by the curing process, it will form 'stiff skin' on the top, which is a concern to create bilayer structure, did author note this phenomenon?

The responses to my comments 1.2 and 1.3 didn't really hit the point, I would recommend author to check and explain a clear answer.

The research on the capacitance based sensing and advanced fabrication of dedicated sensing structures have been well studied. The scientific understanding is somehow inadequate, in terms of clarifying the fundamental structure-property relationship. Several relevant references [see, Mater. Horiz., 2023, DOI: 10.1039/D3MH00056G; Adv Compos Hybrid Mater. 2022, 5, 1537–1547. <https://doi.org/10.1007/s42114-022-00430-5>; Advanced Functional Materials, 2023, <https://doi.org/10.1002/adfm.202301117>] might be good references to expand the scope, but not limited here.

In conclusion, the revision has brought some improvements to the manuscript. However, more clarifications will be needed to make the manuscript reach the threshold for publishing in Nat. Comm. I would recommend a major revision.

Reviewer #2 (Remarks to the Author):

I'm pleased to see that authors addressed all the comments appropriately. My remaining minor comment is that Figure 4 caption should have details of cyclic conditions such as "a cyclic shear test with a maximum shear strain of 66.7% (Figure 4h)" and "cyclic torsional test with a maximum twist angle of 30° (Figure 4i)".

Reviewer #3 (Remarks to the Author):

The study focuses on addressing the limitations of existing ionotronic sensors by developing a new fabrication approach and material design. This approach allowed them to create sensors capable of sensing tension, compression, shear, and torsion, with customizable sensitivities achieved by programming the device architectures. The use of leakage-free polyelectrolyte elastomers in multi-material 3D printing opens new possibilities for manufacturing stretchable ionotronics. By addressing the limitations of stability and functionality simultaneously, this study represents an important advancement in the field of ionotronic sensor technology.

1. The intrinsic properties of the elastomers, including mechanical strength and electrical conductivity, do not demonstrate notable advantages. Similar studies addressing these properties have also been observed frequently in the literature. This reviewer firmly believes that the significance of developing diverse novel applications lies in the utilization of base materials with exceptional performance.
2. The relevant applications of a wearable remote-control unit should be further described in detail, including the design of printed circuit boards and the analysis of multi-channel data acquisition.
3. In the video demonstration, the sensors are adhered to the surface of a rubber glove. Further characterization of the adhesive strength of the devices on surfaces such as skin and fabric can be conducted to demonstrate their potential as wearable devices. Additionally, considerations such as skin-friendliness, breathability, and flexibility of the adhesive materials should be taken into account to ensure user comfort and long-term wearability.
4. Similarly, the deformation of ion-conductive materials can result in changes in electrical resistance. The question arises whether these resistance variations can impact the capacitance response characteristics of the devices. In general, when the resistance value undergoes variations, the capacitance value can also change accordingly.
5. When collecting data from sensors with multiple channels, the possibility of signal crosstalk does exist. Signal crosstalk refers to the interference or coupling of signals between different channels, which can affect the accuracy of data acquisition. How to avoid signal crosstalk in separation of multiple strain information?

REVIEWER COMMENTS

Response: The authors would like to thank the editors and reviewers for their valuable comments which help us further improve the quality of our paper. We have carefully addressed the reviewers' comments and suggestions point-by-point and revised our paper accordingly. Our responses to each of the comments are as follows.

Reviewer #1:

General Comment: The authors have tried to address my previous concerns; however, considerable gap remains from the scientific perspective. The author performed more characterizations. The key is to bring a clear scientific interpretation, rather to pile up the data.

Response: We thank the reviewer for the comments. We have added more in-depth results and discussions, such as finite element analysis (FEA), to enhance scientific interpretations.

Comment 1.1: Regarding to my first query, a mechanics study is mandatory to explain the sensing function of a single mode of deformation such as compression or tension. The current interpretation is far from enough. A FEA analysis can do the job, I suppose. I have second queries on the linearity of plots in Fig. R5 and Fig. R6. The trends appear to me that they are more link segment of a nonlinear curve, the current fit seems very rough.

Response: We thank the reviewer for the constructive suggestions. We have performed FEA for a more in-depth understanding of the sensing functions of the tensile sensor using the COMSOL Multiphysics.

As shown in Figure R1, we model the tensile sensor using the explicit geometries. Because the two ends of the dumbbell-shaped sample have larger cross-sectional areas and are glued to two acrylate plates for clamping, the majority of tensile deformation

occurs in the central segment under uniaxial tension. During deformation, the area increases and the thickness shrinks such that the capacitance increases. Assume the materials to be incompressible, the original length, width, and thickness of the central segment to be l_0 , w_0 , and t_0 . When the length is strained to $(1 + \varepsilon)l_0$ (ε is the tensile strain), due to the Poisson's effect, the width and the thickness become $w_0/\sqrt{1 + \varepsilon}$ and $t_0/\sqrt{1 + \varepsilon}$, respectively. Consequently, $C \propto \frac{(1+\varepsilon)l_0 \times w_0/\sqrt{1+\varepsilon}}{t_0/\sqrt{1+\varepsilon}} \propto \varepsilon$ and $\Delta C/C_0$ is linearly proportional to the tensile strain. The FEA result shows that $\Delta C/C_0$ varies with tensile strain mostly in a linear manner, which is in satisfactory agreement with the experiment, as shown in Figure R1b. The small deviation should be due to the discrepancies in the specific deformation, boundary conditions, and material properties, such as the compressibility of the materials, between FEA and the experiment. In addition, the distributions of strain, capacitance, and electric field before and after deformation have been shown.

We have added Figure R1 as Figure S15 and the following sentences on page 14 in the revised Supplementary Materials: "Finite element analysis (FEA) reveals that, whereas the majority of deformation occurs in the central segment, $\Delta C/C_0$ varies with tensile strain mostly in a linear manner, which is in satisfactory agreement with the experiment (Figure S15)."

Figure R1 (Figure S15 in Supplementary Materials). FEA results of the tensile sensor. (a) Geometries, boundary conditions, and loading conditions of the tensile sensor. (b) Variations of $\Delta C/C_0$ with tensile strain. Distributions of strain, capacitance, and electric field the sensor before (c) and after (d, e, f) deformation.

Furthermore, we have also performed FEA for a more in-depth understanding of the sensing functions of the integrated tensile and compressive sensor. Recall that we have rationally designed the sensor structure such that the two independent electrodes of the sensor are separated much farther than the thickness of the DE layer to minimize signal cross-talks. The DE layer of the tensile sensor is much thinner than the DE layer of the compressive sensor so the relative thickness change of the DE layer of the compressive

sensor is negligible when the tensile sensor is activated. As shown in Figure R2, FEA results confirm that the compressive unit deforms substantially whereas the tensile unit deforms negligibly when the sensor is under compression. Specifically, for a normal compressive strain of 25%, the principal strain of the compressive unit is ~ 0.328 and the principal strain of the tensile unit is only ~ 0.00873 . Figure R2b shows the variation of $\Delta C/C_0$ with compressive strain for capacitance meters C_1 and C_2 . The capacitance of C_1 increases while the capacitance of C_2 barely changes with compressive strain, which is consistent with experimental results. The distributions of principal strain, potential, and electric field, as well as the capacitances of C_1 and C_2 before and after deformation have been shown in Figure R2c-2l. We have added Figure R2 as Figure S22 in the revised Supplementary Materials.

Figure R2 (Figure S22 in Supplementary Materials). FEA results of the integrated tensile and compressive sensor under compression. (a) Geometries, boundary conditions, and loading conditions of the integrated tensile and compressive sensor under compression. (b) $\Delta C/C_0$ varies with compressive strain for capacitance meters C_1 and C_2 . Principal strain field distributions at undeformed state (c) and at a compressive strain of 25% (d). Potential field distributions at undeformed state (e) and at a compressive strain of 25% (f). Electric field distributions at undeformed state (g) and at a compressive strain of 25% (h). The capacitances of C_1 at undeformed state (i) and at a compressive strain of 25% (j). The capacitances of C_2 at undeformed state (k) and at a compressive strain of 25% (l).

As shown in Figures R3, FEA results confirm that the tensile unit deforms substantially whereas the compressive unit deforms negligibly when the sensor is under tension. Specifically, for a normal tensile strain of 100%, the principal strain of the tensile unit is ~ 0.483 and the principal strain of the compressive unit is only ~ 0.0029 . Figure R3b shows the variation of $\Delta C/C_0$ with tensile strain for capacitance meters C_1 and C_2 . The capacitance of C_2 decreases while the capacitance of C_1 barely changes with tensile strain, which is consistent with experimental results. The distributions of principal strain, potential, and electric field, as well as the capacitances of C_1 and C_2 before and after deformation have been shown in Figure R3c-3l. We have added Figure R3 as Figure S23 in the revised Supplementary Materials.

Figure R3 (Figure S23 in Supplementary Materials). FEA results of the integrated tensile and compressive sensor under tension. (a) Geometries, boundary conditions,

and loading conditions of the integrated tensile and compressive sensor under tension. (b) $\Delta C/C_0$ varies with tensile strain for capacitance meters C_1 and C_2 . Principal strain field distributions at undeformed state (c) and at a tensile strain of 100% (d). Potential field distributions at undeformed state (e) and at a tensile strain of 100% (f). Electric field distributions at undeformed state (g) and at a tensile strain of 100% (h). The capacitances of C_1 at undeformed state (i) and at a tensile strain of 100% (j). The capacitances of C_2 at undeformed state (k) and at a tensile strain of 100% (l).

Moreover, we purposely simulate the responses of a badly designed sensor, i.e. the two independent electrodes of the sensor are relatively close to each other and the DE layer of the tensile sensor is even thicker than the DE layer of the compressive sensor, for comparison. The dimensions of the badly designed sensor are shown in Figure R4a. FEA results confirm that the global tensile deformation of the sensor not only causes the tensile unit to deform substantially but also causes the compressive unit to deform somewhat, especially for the adjacent regions. Specifically, for a normal tensile strain of 50%, the principal strain of the tensile unit is ~ 0.384 and the principal strain of the compressive unit reaches ~ 0.0375 . Figure R4b shows that both the capacitances of C_1 and C_2 change notably with tensile strain. The bending deformation at the central region is due to the asymmetric stress distribution that the stress at the bottom is larger than the stress at the top, resulting in a bending moment. The distributions of principal strain, potential, and electric field, as well as the capacitances of C_1 and C_2 before and after deformation have been shown in Figure R4c-4l. We have added Figure R4 as Figure S24 in the revised Supplementary Materials.

Figure R4 (Figure S24 in Supplementary Materials). FEA results of a bad design of the integrated tensile and compressive sensor under tension. (a) Geometries,

boundary conditions, and loading conditions of the badly-designed integrated tensile and compressive sensor under tension. (b) $\Delta C/C_0$ varies with tensile strain for capacitance meters C_1 and C_2 . Principal strain field distributions at undeformed state (c) and at a tensile strain of 50% (d). Potential field distributions at undeformed state (e) and at a tensile strain of 60% (f). Electric field distributions at undeformed state (g) and at a tensile strain of 50% (h). The capacitances of C_1 at undeformed state (i) and at a tensile strain of 50% (j). The capacitances of C_2 at undeformed state (k) and at a tensile strain of 50% (l).

We have added the following sentences on page 18 in the revised manuscript: “FEA results also show prominent differences between the signals of C_1 and C_2 when the sensor is subject to compression (Figure S22) or tension (Figure S23). The key point of signal decoupling is to minimize the associated deformation of one sensor when another sensor is deformed through appropriate structure design. As a counterexample, both the capacitances of C_1 and C_2 of a badly designed sensor change notably with tensile strain (Figure S24).”

In addition, we have added a section entitled “Finite element analysis” to the Method section in the revised manuscript:

“FEA was performed using the commercial package COMSOL Multiphysics 6.0. According to the experimental data, both the PEE and DE were modeled as incompressible neo-Hookean materials with shear moduli of 90.7 kPa and 294 kPa, respectively. DE was modeled as a linear dielectric material with relative dielectric constant $\epsilon_r = 3$, and PEE was simplified as an equipotential body. The deformation caused by electrostatic force was ignored.

Tensile sensor. The upper PEE was applied with a voltage of 1 V and the lower PEE was grounded. For boundary conditions, the left side of the sensor was fixed and the right side was subjected to a specified displacement L . The maximum value of the specified displacement L was set to be 10 mm.

Integrated tensile and compressive sensor. The bottom-right PEE was grounded and both the top-right and bottom-left PEEs were applied with 1 V. For compression, the bottom-right PEE was fixed and the top-right PEE was compressed by a displacement L_1 . For tension, the bottom-right PEE was fixed and the bottom-left PEE was elongated by a displacement L_2 . L_1 and L_2 are 0.75 mm and 1 mm for the rationally designed sensor, and L_2 is 2.5 mm for the badly designed sensor.”

As for the linearity of the plots in Fig. R5 and Fig. R6, i.e. the $\Delta C/C_0$ versus shear strain curves and the $\Delta C/C_0$ versus twist angle curves, we would like to clarify that the three $\Delta C/C_0$ versus shear strain curves are not all linear. For a flat front-line design, the curve is linear because $\Delta C \propto \Delta l$ (red curve). For a zigzag pattern design, the curve is quadratic because $\Delta C \propto (\Delta l)^2$ (green curve). For a parabolic pattern design, the curve is cubic because $\Delta C \propto (\Delta l)^3$ (blue curve). The R^2 values of the three fittings are 0.99916, 0.99998, and 0.99998, respectively. For the $\Delta C/C_0$ versus twist angle curves, the fittings indeed deviate from the experimental data, especially at small twist angles. The reasons are as follows. First, the difference between the bisection and the quartering designs is theoretically small at small twist angles. Second, the absolute capacitances of the two types of sensors are small, on the order of 1 pF, due to the large distance between the two electrodes and the presence of air, so the signal-to-noise ratio is low and the measured capacitance of the sensor can be profoundly affected by noise. Third, we have used a self-built LabVIEW-controlled torsional loading system to apply torsion to the sensor and the lack of controlling accuracy and stability might cause additional noise to the measured capacitance. Nevertheless, the linear fittings are satisfactory when the twist angle is larger than $\sim 10^\circ$. The R^2 values of the two fittings are 0.99641 and 0.99273, respectively. Better fittings could be achieved through further optimizations by, e.g. reducing the distance between the two electrodes and using a more reliable loading systems, but these are out of the scope of current work.

Comment 1.2: For the photo-curing assessment, there is a concern on the gradient caused by the curing process, it will form ‘stiff skin’ on the top, which is a concern to create bilayer structure, did author note this phenomenon?

Response: We appreciate the reviewer for this insight. For 3D-printed thin laminated structures, the stiffness gradient might cause the bending of the structures due to residual stress or inhomogeneous swelling. However, each printed layer only has a thickness of $\sim 50 \mu\text{m}$ while the overall thickness of the printed sensors is on the order of 1 mm in our experiments. Note that the bending stiffness is proportional to the thickness to the third power, $EI \propto H^3$, where E is Young’s modulus, I is the moment of inertial of the cross-section, and H is thickness. As a result, the stiffness gradient barely affects the properties of our printed sensors. Furthermore, we purposely placed the printed structure in a UV oven for 1 hour for complete curing of the printed elastomers to alleviate the influences of gradient curing. Both the thicknesses of each printed layer and the sensor and the post-curing treatment have been stated in the Method section of the manuscript: “...and the layer thickness was $50 \mu\text{m}$.”, “After printing, we used ethanol to rinse the printed structure and remove the uncured precursor. Subsequently, we placed the printed structure in a UV oven with 365 nm wavelength for 1 hour for complete curing of the elastomers.”

Comment 1.3: The responses to my comments 1.2 and 1.3 didn’t really hit the point, I would recommend author to check and explain a clear answer.

Response: We thank the reviewer for the reminder. We have carefully re-checked the previous comments 1.2 and 1.3.

We have addressed previous comments 1.2.1 and 1.2.3 in the last response. As for previous comment 1.2.2, “*In Fig1f, the testing method needs a diagram to illustrate the process with defined parameters;*”, we have added Figure 3e and Figure 3f in the last response to compare different failure modes of printed sample and assembled sample but without giving a diagram to illustrate the 180° peeling test with defined parameters. As shown in Figure R5, we now add a schematic with defined parameters to illustrate

the 180° peeling test and add it to the revised Supplementary Materials as Figure S12. The corresponding sentence on page 12 has been rewritten as: “The adhesion energies, given by the plateaued normalized force, $2F_{ss}/W$, where F_{ss} is the steady-state peel force and W is the width of the sample (Figure S12), are 339.3 J m⁻² and 4.1 J m⁻² for printed and assembled samples, respectively.”

Figure R5 (Figure S12 in Supplementary Materials). A schematic diagram of 180° peeling tests. The width of the sample is w and the total thickness of the sensor is d . The two arms of the sample are pulled by a pair of forces F . Two soft but inextensible backing layers are glued to the top and bottom surfaces to prevent elongation of the two arms so that, at the steady state, the work done by the peel force is totally converted into fracture energy.

For previous comment 1.3:

“*The demonstration is lack of lots of quantitated details, the responding time has not been well examined.*”, in the last response, we conducted a response time test and added Figure S22 (now Figure S28) in the Supplementary Materials, as shown in Figure R6.

Figure R6 (Figure S28 in Supplementary Materials). Response time tests of (a) shear sensor and (b) compressive sensor of the remote-control unit.

“The mechanical input based signal generation seems very slim, not mention that the other electronic units are commercial based. A lot of capacitive based Human-Machine Interface can be demonstrated, the current demonstration seems not the best to show the advantage for this work.” The key point of our work is to resolve the deficiencies in sensing mode and sensing stability in ionotronic sensing simultaneously, by synthesizing a new type of photo-curable leakage-free polyelectrolyte elastomer and using the DLP-based multi-material 3D printing technique to fabricate a variety of long-term stable ionotronic sensors with multi-mode sensing capabilities. Sensors that can sense mechanical stimuli are important for, e.g. biological systems to perceive and interact with the surroundings for adaption and survival and robots for accurate manipulation and safe human-machine interactions. We have demonstrated diverse ionotronic sensors capable of sensing various mechanical stimuli, including tension, compression, shear, and torsion, as well as combined tension and compression, combined compression and shear, and combined compression and torsion without signal cross-talks. In our work, we focus on the materials, mechanics, manufacturing, and performances of the sensors within the paradigm of ionotronics, and use commercial products for other electronic units.

We totally agree with the reviewer that a lot of capacitive-type human-machine interfaces can be demonstrated. Recall that one of the advantages of our work is the ability to fabricate a variety of ionotronic sensors with multi-mode sensing capabilities with the printable leakage-free polyelectrolyte elastomer and the DLP-based multi-material 3D printing technique. In this sense, our demonstration shows the advantages well with the following reasons. First, the remote-control sensor of the demonstration is an integrated sensor that can sense compression and shear. Second, the spatial layout of the four shear sensors and the compressive sensor are rationalized such that the five sensing channels can be well decoupled with mitigated signal cross-talks. Third, the sensing mechanism of the shear sensor is different from the previous one (Figure 4e) and is purposely designed to behave like a switch. As shown in Figure R7, the sensor accommodates a capacitor due to the air in series with two capacitors due to the electric double layer. Upon shear, the PEEs come into contact with each other to eliminate the air capacitor, resulting in a giant capacitance change by orders of magnitude. Experimental results show that the capacitance changes by more than 10^4 times upon stimulation. Such a giant signal is noise-tolerant and highly beneficial for circuit design. Note that the feasibility of achieving the above-mentioned functionalities of the remote-control sensor is attributed to the high flexibility in the structure design enabled by the DLP-based multi-material 3D printing technique. Therefore, whereas many capacitive-type human-machine interfaces can be demonstrated, the printed wearable wireless remote-control unit for a drone shows the advantage of our work well.

Figure R7 (Figures 6b and 6c in the manuscript). Shear sensor of the remote-control unit. (a) Schematics of the operation modes of the shear sensors. (b) Working mechanism of the shear sensor.

Comment 1.4: The research on the capacitance based sensing and advanced fabrication of dedicated sensing structures have been well studied. The scientific understanding is somehow inadequate, in terms of clarifying the fundamental structure-property relationship. Several relevant references [see, Mater. Horiz., 2023, DOI: 10.1039/D3MH00056G; Adv Compos Hybrid Mater. 2022, 5, 1537–1547. <https://doi.org/10.1007/s42114-022-00430-5>; Advanced Functional Materials, 2023, <https://doi.org/10.1002/adfm.202301117>] might be good references to expand the scope, but not limited here.

Response: We thank the reviewer for suggesting the latest works. We have rewritten relevant sentences to expand the scope of the Introduction for the discussion of other capacitive type sensors and cited the suggested papers at suitable positions, i.e. Ref. 21, 22, and Ref. 40, in the revised manuscript.

In the second paragraph of the Introduction: “Ionotronic sensors are emerging for tactile perception. Compared to traditional sensors entirely based on electronic materials^{21,22}, ionotronic sensors feature extraordinary softness, flexibility/stretchability, and optical transparency.”

In the first paragraph on page 12: “Cohesive rupture occurs during the peeling of the printed bilayer that PEE residues are left on the surface of DE after peeling (Figure 3e), meaning that the interface is tougher than the bulk of PEE⁴⁰.”

Comment 1.5: In conclusion, the revision has brought some improvements to the manuscript. However, more clarifications will be needed to make the manuscript reach the threshold for publishing in Nat. Comm. I would recommend a major revision.

Response: We thank the reviewer for the positive remarks on our revisions. We have revised the manuscript according to the comments. Revisions include FEA results for the tensile sensor (Figure S15), the integrated tensile and compressive sensor under compression (Figure S22) or tension (Figure S23), and the badly-designed integrated tensile and compressive sensor under tension (Figure S24), a diagram to illustrate the 180° peeling test with defined parameters (Figure S12). We wish the reviewer has satisfied with the current revision.

Reviewer #2:

Comment 2.1: I’m pleased to see that authors addressed all the comments appropriately. My remaining minor comment is that Figure 4 caption should have details of cyclic conditions such as “a cyclic shear test with a maximum shear strain of 66.7% (Figure 4h)” and “cyclic torsional test with a maximum twist angle of 30° (Figure 4i)”.

Response: We thank the reviewer for the positive remarks on our revisions and the kind suggestions. We have revised the captions of Figure 4h and Figure 4i as follows: “(h) Cyclic shear test with a maximum shear strain of 66.7%.”, “(i) Cyclic torsional test of the bisected torsional sensor with a maximum twist angle of 30°.”.

Reviewer #3:

General Comment: The study focuses on addressing the limitations of existing ionotronic sensors by developing a new fabrication approach and material design. This approach allowed them to create sensors capable of sensing tension, compression, shear, and torsion, with customizable sensitivities achieved by programming the device architectures. The use of leakage-free polyelectrolyte elastomers in multi-material 3D printing opens new possibilities for manufacturing stretchable ionotronics. By addressing the limitations of stability and functionality simultaneously, this study represents an important advancement in the field of ionotronic sensor technology.

Response: We thank the reviewer for the positive remarks on our work.

Comment 3.1: The intrinsic properties of the elastomers, including mechanical strength and electrical conductivity, do not demonstrate notable advantages. Similar studies addressing these properties have also been observed frequently in the literature. This reviewer firmly believes that the significance of developing diverse novel applications lies in the utilization of base materials with exceptional performance.

Response: We would like to emphasize that the key point of our work is to resolve the deficiencies in sensing mode and stability, two pervasive but vital limitations for ionotronic sensing. We do so by synthesizing a new type of **leakage-free polyelectrolyte elastomer** and using the **DLP-based multi-material 3D printing** technique to fabricate a variety of long-term stable ionotronic sensors with multi-mode sensing capabilities.

For the materials, we totally agree with the reviewer that significant development of applications relies on the utilization of base materials with exceptional performances. On one hand, the polyelectrolyte elastomer has to be solvent-free to avoid solvent leakage. On the other hand, at least one type of ion should be fixed to the polymer network to avoid ion leakage. However, fixing ions to the polymer network inevitably restricts the mobility of ions and thus reduces the ionic conductivity. As a result,

polyelectrolyte elastomers intrinsically possess relatively low ionic conductivity (typically 10^{-5} - 10^{-3} S m⁻¹), lower than that of gel-based ionic conductors such as ionic hydrogels and ionogels (10^{-2} - 10^1 S m⁻¹) by orders of magnitude. Our newly designed and optimized polyelectrolyte elastomer, p(BS-co-MEA), exhibits balanced mechanical and electrical properties, enabling a variety of long-term stable iontronic sensors with multi-mode sensing capabilities. Compared to other polyelectrolyte elastomers, e.g. the recent ones reported by Kim et al. in 2020 in *Science* (Ref. 17 in the manuscript), our p(BS-co-MEA) shows comparable mechanical and electrical properties. We compare the conductivity, elongation at break, and Young's modulus of the two materials in Table R1.

Table R1. A comparison of our PEE and the PEE reported in Ref. R1.

	Our PEE	Other PEE ^{Ref.R1}
Conductivity (S m ⁻¹)	10^{-3} - 10^{-2}	10^{-4} - 10^{-3}
Elongations at break	170%	120%, 140%
Young's modulus (kPa)	256	100

Ref. R1. Kim, H. J., Chen, B., Suo, Z. & Hayward, R. C. Ionoelastomer junctions between polymer networks of fixed anions and cations. *Science* **367**, 773–776 (2020).

We agree that designing and synthesizing polyelectrolyte elastomers of exceptional performances will be beneficial for high-performance iontronic sensors, which requires additional studies but is beyond the scope of current work.

Comment 3.2: The relevant applications of a wearable remote-control unit should be further described in detail, including the design of printed circuit boards and the analysis of multi-channel data acquisition.

Response: We thank the reviewer for the constructive suggestions.

First, for more clarity, we add a digital image and two schematics to show the detailed geometrical information of the remote-control sensor, as shown in Figure R8. We have added the figure to the revised Supplementary Materials as Figure S26. The

corresponding sentence on page 21 has been rewritten as: “The remote-control unit integrates five sensors: one compressive sensor and four shear sensors (Figure S26), which are used as the input ports.”

Figure R8 (Figure S26 in Supplementary Materials). Remote-control sensor. (a) Photograph of the remote-control sensor with copper wires connected. Scale bar: 10 mm. (b) Schematic of the remote-control sensor with relevant dimensions indicated. (c) Schematic of the cutaway view with relevant dimensions indicated. The unit is mm.

Second, we have rewritten the “Remote-control system for a drone” section in the Method section to add more detailed descriptions about the design of the printed circuit boards and the analysis of multi-channel data acquisition in the revised manuscript as follows: “A remote-control unit consisting of one compressive sensor and four shear sensors was designed and 3D printed. A copper wire was inserted into the cylindrical PEE(C₅) during the printing process. Four copper wires were fixed to each side of the remote-control unit with PEE precursor. The five sensors of the remote-control unit shared the same ground electrode and were separately connected to five channels of a multiple relay, which was then connected to an LCR meter (TH2838A, Changzhou Tonghui Electronic Co. Ltd, China). The relay received and processed one signal once at a time and its on-off state was regulated by the digital signals sent from an Arduino UNO board. The output end of the Arduino UNO board was connected to five channels of another multiple relay, which was further connected to the corresponding pins of the PCB of the drone controller by welding. The LCR meter operated at medium speed with a frequency of 1 kHz and a voltage of 0.5 V. The real-time capacitances and the normalized capacitances of the five sensors were measured in a loop one after another by an LCR meter and were recorded by a LabVIEW program (National Instruments, Austin, TX, USA), and were displayed on the computer screen. After the program got

started, the capacitance of each sensor was measured 20 times and then the average value was taken as the initial capacitance. Subsequently, the capacitance increased upon the loading of a finger. Once the normalized capacitance C/C_0 of a sensor was larger than the prescribed threshold, the corresponding pin was switched on and a corresponding signal was generated, which eventually lead to an operation command for the drone. During operation, the remote-control unit was worn on the hand back and connected to a controlling circuit containing two relays, an LCR meter, a LabVIEW controlling program, and an Arduino board (Figure S30). In response to the perturbation of a finger, the capacitance of a sensor increased. When the value of C/C_0 exceeded a threshold, the circuit was switched on and a corresponding steering order was sent to the drone. The thresholds for capacitors C_1 , C_2 , C_3 , and C_4 were set to be 1000 and the threshold for C_5 was set to be 1.1. The order of “flip” was a pre-order that the signal of C_5 should be followed by another order of one of the other four sensors, such that the drone would flip in the corresponding direction. The drone executed the orders accurately. The source file of the LabVIEW program used to control the remote-control system was uploaded as a supplementary material.”

The LabVIEW program for the remote-control system is shown in Figure R9 and has been added to the revised Supplementary Materials as Figure S27. The corresponding sentence on page 21 has been rewritten as: “A customized LabVIEW controlling program collects and processes the signal and sends it to a printed circuit board (PCB), which further generates a command to the drone via electromagnetic waves (Figure S27).” In addition, the source file of the LabVIEW program has been uploaded as a supplementary material.

Figure R9 (Figure S27 in Supplementary Materials). The LabVIEW program for the remote-control system. The program runs from left to right. First, set the parameters for capacitance measurement and calibrate the initial value of each capacitance channel. Then, assign one channel of the relays to each capacitor after entering the loop structure of capacitance acquisition. When the measurement starts, close the circuit of one channel at a time meanwhile keeping other channels open, measure the capacitance of the connected channel, and then disconnect the channel. Repeat the process for five channels. After that, enter the drone control circuit and compare the five capacitance values with the initial values. Finally, perform a closed-circuit operation to generate an action command to the drone based on the corresponding relay channel.

Comment 3.3: In the video demonstration, the sensors are adhered to the surface of a rubber glove. Further characterization of the adhesive strength of the devices on surfaces such as skin and fabric can be conducted to demonstrate their potential as wearable devices. Additionally, considerations such as skin-friendliness, breathability, and flexibility of the adhesive materials should be taken into account to ensure user comfort and long-term wearability.

Response: We thank the reviewer for the kind suggestions. Indeed, adhesion between the printed sensor and other materials is an important issue in practical applications, but

developing a new type of adhesive material and investigating its performances such as skin-friendliness, breathability, and flexibility deserve an independent project and are beyond the scope of current work. However, just as the robust adhesion achieved between the PEE and tango in the printed sensors, achieving robust adhesion between the sensor and other materials is feasible using the DLP-based 3D printing. For example, we use a layer of polyacrylamide hydrogel to adhere a printed sensor to various materials, including fabric, skin, plastic, and metal (aluminum alloy), as shown in Figure R10a. We further perform a 180° peeling test to probe the adhesion between a printed sensor and a non-woven fabric (Figure R10b). Cohesive failure occurs along the hydrogel layer (Figure R10c), which is an indicator of strong adhesion.

We add Figure R10 as Figure S28 in the revised Supplementary Materials and add the following sentences on page 22 in the revised manuscript: “Furthermore, the adhesion between the remote-control unit and the substrate is important in practical deployments. Similar to the robust adhesion achieved between the PEE and DE in the printed sensors, achieving robust adhesion between the sensor and other materials is feasible using the DLP-based 3D printing. As an example, we print a layer of polyacrylamide hydrogel to strongly adhere a printed sensor to various materials, including fabric, skin, plastic, and metal (Figure S28).”

We also add a section, entitled “Adhering a printed sensor and a fabric using a hydrogel adhesive” in the Methods section: “A hydrogel solution was prepared using acrylamide (AAM) as the monomer, 0.625 mol% PEGDA as the crosslinker and 5 wt% TPO as the photo-initiator. AAM, PEGDA, and TPO were dissolved in deionized water with a water content of 80 wt% to form a transparent precursor as the printing ink. We first printed the sensor with a dimension of 50 mm × 10 mm × 1.5 mm (0.5 mm for each layer) following the same steps as before. Then we printed a layer of the hydrogel with a thickness of 1 mm as an adhesive layer on the sensor. After printing, the printed structure was subjected to UV light irradiation with 365 nm wavelength for 1 h for complete curing of the printed structure. For the 180° peeling test, a fabric was bonded onto the

hydrogel and the test was performed on an Instron 5966 with a 100N load cell at a speed of 10 mm min^{-1} .”

Figure R10 (Figure S28 in Supplementary Materials). Adhesion between the printed sensor and various substrates using a layer of hydrogel adhesive. (a) Photographs of a 3D-printed sensor adhered to the skin, plastic (acrylate sheet), and metal (aluminum alloy). Scale bars: 10mm. **(b)** A schematic diagram of the 180° peeling test between a printed sensor and a non-woven fabric adhered by a layer of polyacrylamide hydrogel. **(c)** 180° peeling curve. The inset shows the cohesive failure of the sample.

In addition, we would like to point out that the adhesion of soft materials and soft devices is an emerging subject of multi-disciplines such as mechanics, chemistry, topology, etc. Hydrogels as soft and wet adhesives for interfacing human beings and soft machines, as well as other types of soft adhesive materials, have been extensively

investigated in recent years. Here we have only demonstrated the feasibility of achieving robust adhesion for the printed sensors using one type of hydrogel. A more comprehensive and systematic study of new adhesive materials requires additional investigations but is outside the scope of this work.

Comment 3.4: Similarly, the deformation of ion-conductive materials can result in changes in electrical resistance. The question arises whether these resistance variations can impact the capacitance response characteristics of the devices. In general, when the resistance value undergoes variations, the capacitance value can also change accordingly.

Response: We appreciate the reviewer for this insight. Both resistance and capacitance will change when the device is deformed. For ideal dielectrics and ideal conductors, both changes are the consequences of geometrical changes. Therefore, we agree with the reviewer that the resistance change of the polyelectrolyte elastomer will affect the response characteristics of the sensor, e.g. the charging and discharging time of the capacitor, the response speed, or the RC delay of the circuit. However, the resistance change barely alters the capacitance value of our sensor. As shown in Figure R11, we perform the following FEA for verification.

Figure R11. The capacitance of a parallel-plate capacitor with electrodes of different resistance values. (a) Schematic of the parallel-plate capacitor with relevant information indicated. (b) Equivalent circuit of the capacitor. (c) The capacitances of the capacitor with three resistance values of the electrodes.

We model the capacitive responses of a 10 mm wide parallel-plate capacitor, consisting of a layer of 2 mm thick dielectric elastomer sandwiched between two layers of 1 mm thick conductive elastomer, as shown in Figure R11a. The dielectric elastomer has a conductivity of $\sigma = 10^{-12} S/m$ and a relative dielectric constant of $\epsilon_r = 3$. The conductor elastomer has a conductivity of $\sigma = 10^{-2} S/m$ and a relative dielectric constant of $\epsilon_r = 30$. The top electrode is applied with 1 V and the bottom electrode is grounded. We model each layer as a resistor in parallel with a capacitor (Figure R11b). The subscript “1” represents the conductive elastomer and the subscript “2” represents the dielectric elastomer. Without losing generality, we select three resistance values for R_1 , conduct FEA, and calculate the capacitance. As shown in Figure R11c, the

capacitances are almost the same whereas the resistance value expands over three orders of magnitude.

The details about the modeling are as follows. Introduce the current conservation equation and make the material comply with Ohm's constitutive law. The conductivity and relative dielectric constant were set as 10^{-12} S/m and 3 for dielectric elastomer, and 10^{-2} S/m and 30 for conductive elastomer. Calculate the impedance using the current obtained from FEA, and then use the imaginary part of the impedance to calculate the capacitance value based on the selected circuit model.

Comment 3.5: When collecting data from sensors with multiple channels, the possibility of signal crosstalk does exist. Signal crosstalk refers to the interference or coupling of signals between different channels, which can affect the accuracy of data acquisition. How to avoid signal crosstalk in separation of multiple strain information?

Response: We appreciate the reviewer for this insight. Indeed, signal crosstalk is an important issue that needs to be carefully resolved when designing sensors with multi-mode sensing capabilities. The signals of different channels might interfere with each other. Since the capacitance change mainly depends on the geometrical change, the key to avoiding signal crosstalk is to minimize the associated deformation of other sensors when deforming one sensor through appropriate structure design. Thanks to the high flexibility in the structure design of multi-material 3D printing, we can design and fabricate integrated ionotronic sensors that can sense different stimuli without prominent mutual signal interferences.

As shown in Figure 5, we have fabricated three types of integrated ionotronic sensors. Take the integrated tensile and compressive sensor as an example, we rationally design and fabricate an integrated tensile and compressive sensor that can decipher the signals of compression, tension, or their combination. The design and principle of the sensor are sketched in Figure R12a. The right part constitutes a compressive sensor monitored by the capacitance meter C_1 , and the bottom part constitutes a tensile sensor monitored

by the capacitance meter C_2 . Note that the two sensors have one shared electrode and two independent electrodes, which are separated much farther than the thickness of the DE layer to minimize signal cross-talks. In addition, the DE layer of the tensile sensor is much thinner than the DE layer of the compressive sensor such that, the relative thickness change of the DE layer of the compressive sensor is negligible when the tensile sensor is activated. Specifically, the projected distance between the two independent electrodes is 13 mm, the thickness of the DE layer of the compressive unit is 2 mm, and the DE layer of the tensile unit is 1 mm, as shown in Figure R12b. Experimentally, under 25% compressive strain, C_1 increases by 27.87% and C_2 increases by 2.22%, giving a signal ratio of 12.6; under 50% tensile strain, C_2 decreases by 10.14% while C_1 increases by 0.32%, giving a signal ratio of 31.7.

Figure R12. Integrated tensile and compressive sensor. (a) Design and principle of the integrated tensile and compressive sensor. Capacitance meter C_1 measures compression and capacitance meter C_2 measures tension. (b) Schematics of the top view and side view of the sensor with relevant dimensions indicated. (c) $\Delta C/C_0$ of C_1 and C_2

when the compressive unit or the tensile unit is activated. (d) Principal strain field at undeformed state. (e) Principal strain field at a compressive strain of 25%. (f) $\Delta C/C_0$ varies with compressive strain for capacitance meters C_1 and C_2 . (g) Principal strain field at undeformed state. (h) Principal strain field at a tensile strain of 100%. (i) $\Delta C/C_0$ varies with tensile strain for capacitance meters C_1 and C_2 .

In addition, we have performed FEA and the results validate our design principles well. As shown in Figure R12d & e, for a normal compressive strain of 25%, the principal strain of the compressive unit is ~ 0.328 while the principal strain of the tensile unit is only ~ 0.00873 . Figure R12f shows the variation of $\Delta C/C_0$ with compressive strain for capacitance meters C_1 and C_2 . The capacitance of C_1 increases while the capacitance of C_2 barely changes with compressive strain. As shown in Figure R12g & h, for a normal tensile strain of 100%, the principal strain of the tensile unit is ~ 0.483 while the principal strain of the compressive unit is only ~ 0.0029 . Figure R12i shows the variation of $\Delta C/C_0$ with tensile strain for capacitance meters C_1 and C_2 . The capacitance of C_2 decreases while the capacitance of C_1 barely changes with tensile strain.

REVIEWERS' COMMENTS

Reviewer #1 (Remarks to the Author):

This revision has addressed my concerns adequately, I would recommend to accept for publishing.

Reviewer #3 (Remarks to the Author):

This reviewer is satisfied with the revision of the manuscript. It can be accepted in its present form.

REVIEWER COMMENTS

Response: The authors would like to thank the reviewers for their valuable comments. Our point-by-point responses to each of the comments are as follows.

Reviewer #1:

This revision has addressed my concerns adequately, I would recommend to accept for publishing.

Response: We thank the reviewer for the positive remarks on our revisions and the kind suggestions.

Reviewer #3:

This reviewer is satisfied with the revision of the manuscript. It can be accepted in its present form.

Response: We thank the reviewer for the positive remarks on our revisions and the kind suggestions.